## META-RESEARCH

# Systemic racial disparities in funding rates at the National Science Foundation

**Abstract** Concerns about systemic racism at academic and research institutions have increased over the past decade. Here, we investigate data from the National Science Foundation (NSF), a major funder of research in the United States, and find evidence for pervasive racial disparities. In particular, white principal investigators (PIs) are consistently funded at higher rates than most non-white PIs. Funding rates for white PIs have also been increasing relative to annual overall rates with time. Moreover, disparities occur across all disciplinary directorates within the NSF and are greater for research proposals. The distributions of average external review scores also exhibit systematic offsets based on PI race. Similar patterns have been described in other research funding bodies, suggesting that racial disparities are widespread. The prevalence and persistence of these racial disparities in funding have cascading impacts that perpetuate a cumulative advantage to white PIs across all of science, technology, engineering, and mathematics.

**CHRISTINE YIFENG CHEN\*, SARA S KAHANAMOKU\*, ARADHNA TRIPATI\*, ROSANNA A ALEGADO\*, VERNON R MORRIS\*, KAREN ANDRADE AND JUSTIN HOSBEY**

**\*For correspondence:**
cychen@llnl.gov (CYC);
sara.kahanamoku@berkeley.edu (SSK);
atripati@g.ucla.edu (AT);
rosie.alegado@hawaii.edu (RAA);
vernon.morris@asu.edu (VRM)

## Introduction

Federal science agencies steer and implement national research priorities through grant-making activities, administering funds to hundreds of thousands of researchers at colleges, universities, research institutions, and other organizations across the nation. In 2019, more than half of all research expenditures at US higher education institutions were supported by the federal government through agencies like the National Science Foundation (NSF; $5.3 billion) and the National Institutes of Health (NIH; $24.4 billion) (*National Science Board, 2021b*). As mainstays of the scientific enterprise in the US, these funding bodies exert major influence on the programs and priorities of all higher education and research organizations, conferring economic stability and social capital to individuals and institutions by awarding grants.

In academia, particularly at research-intensive universities, grants underpin every aspect of a researcher's capacity to produce knowledge and innovations. Support for equipment and facilities, stipends and salaries for trainees and personnel, and publication costs all generally depend on grant funding. More funding leads to more research, publications, and reputational prestige, which attracts more talent and generates more research output. Thus, grant awards play a crucial role in research productivity and, by extension, the success and longevity of academic careers.

Funding agencies solicit proposals from principal investigators (PIs) and process them through an evaluation system, making awards based on scientific merit and potential benefit to society. However, several studies over the past decade have revealed inequalities in the allocation of research funding, most notably at the NIH. A 2011 study showed that Black PIs were funded at roughly half the rate as white PIs (*Ginther et al., 2011*). Subsequent analyses surfaced additional inequalities across race (*Ginther et al., 2012*; *Hoppe et al., 2019*; *Erosheva et al., 2020*; *Lauer et al., 2021*; *Ginther et al., 2018*; *Ginther et al., 2016*; *Nikaj et al., 2018*), gender (*Ginther et al., 2016*; *Nikaj et al., 2018*; *Oliveira et al., 2019*), age (*Levitt and Levitt, 2017*), and institution (*Ginther et al., 2012*; *Hoppe et al., 2019*;

*Wahls, 2019*; *Katz and Matter, 2020*). Despite a decade of efforts within NIH to reduce disparities, many remain or have worsened (*Taffe and Gilpin, 2021*; *Lauer and Roychowdhury, 2021*). Such findings are not confined to federal funding bodies: the Wellcome Trust, one of the largest philanthropic funders of scientific research in the world, recently identified similar disparities by race in the distribution of their awards (*Wellcome Trust, 2021*; *Wellcome Trust, 2022*; *Wild, 2022*).

Here we ask whether racial funding disparities are observed at the NSF, the flagship US agency for science, technology, engineering, and mathematics (STEM) research. In contrast to agencies with mission-oriented priorities in biomedicine, space, and energy, NSF has the federal responsibility to support basic research in all areas of STEM, as well as STEM education and workforce development. We examine data on funding rates, award types, and proposal review scores disaggregated by PI race and ethnicity from 1996 to 2019. These data are publicly available in federally mandated annual reports on the NSF merit review process and describe award or decline decisions for over 1 million proposals. Demographic information is collected at the time of proposal submission, when PIs voluntarily provide information on ethnicity — Hispanic or Latino, or not — and race — American Indian or Alaska Native (AI/AN), Asian, Black or African American (Black/AA), Native Hawaiian or other Pacific Islander (NH/PI), and/or white. Because the contents of the merit review reports have evolved over the years, some data are only available for a limited period (e.g., review scores are only available for 2015 and 2016). Nevertheless, we examine all available data to describe and analyze patterns in funding outcomes by PI race and ethnicity.

## Results

### Racial disparities in NSF funding rates

The NSF receives tens of thousands of high-quality submissions each year, many more than it can fund (*Figure 1A*). From 1996 to 2019, the overall funding rate, or the proportion of proposals that were awarded, fluctuated between 22% and 34% due to factors such as changing budgets and proposal submission numbers. For example, stimulus funding in 2009 from the American Recovery and Reinvestment Act raised the funding rate to 32%, whereas a significant increase in proposals in 2010 lowered the funding rate to 23%.

Despite year-to-year variability in the overall funding rate, there are persistent and significant differences in funding rates between proposals submitted by PIs from each racial and ethnic group. To investigate these differences, for each year, we calculated relative funding rates for each group, normalizing by the annual overall funding rate. We find that proposals by white PIs were consistently funded at rates higher than the overall rate, with an average relative funding rate of +8.5% from 1999 to 2019 (*Figure 1B*; *Figure 1—figure supplement 1*). The relative funding rate for proposals by white PIs also steadily increased during this period, from +2.8% in 1999 to +14.3% in 2019. In contrast, proposals by most non-white PIs, specifically Asian, Black/AA, and NH/PI PIs, were consistently funded below the overall rate, with average relative funding rates of –21.2%, –8.1%, and –11.3%, respectively.

The relative contributions of proposals by PIs from each group remain unchanged despite shifts in the number of proposals submitted by each group over time. Submissions by white PIs comprise the majority of proposals throughout the study period (*Figure 1—figure supplement 2*). In 2019, the competitive pool of proposals included 20,400 submissions by white PIs (66% among proposals from PIs who identified their race); 9,241 by Asian PIs (29%); 1,549 by Hispanic or Latino PIs (5%); 929 by Black/AA PIs (3%); 99 by AI/AN PIs (0.3%); and 47 by NH/PI PIs (0.2%) (*Figure 2*). Groups with fewer proposals experienced the greatest year-to-year variability in relative funding rates.

These persistent funding rate disparities are realized as large differences in the absolute number of proposals awarded to PIs in each group. For example, of the 41,024 proposals considered in 2019, the NSF selected 11,243 for funding, or 27.4%. Proposals by white PIs were funded above this overall rate at 31.3%, yielding 6,389 awards (*Figure 2*). If proposals by white PIs had been funded instead at the overall rate of 27.4%, only 5,591 proposals would have been awarded. Thus, an "award surplus" of 798 awards was made to white PIs above the overall funding rate in 2019. In contrast, proposals submitted by the next largest racial group, Asian PIs, were funded at a 22.7% rate, yielding 2,073 awards. If the funding rate for proposals by Asian PIs had been equal to the overall rate, one would instead expect 2,505 awards, or 432 additional awards. We refer to the number of awards required to bridge such gaps in funding rate as the "award deficit."

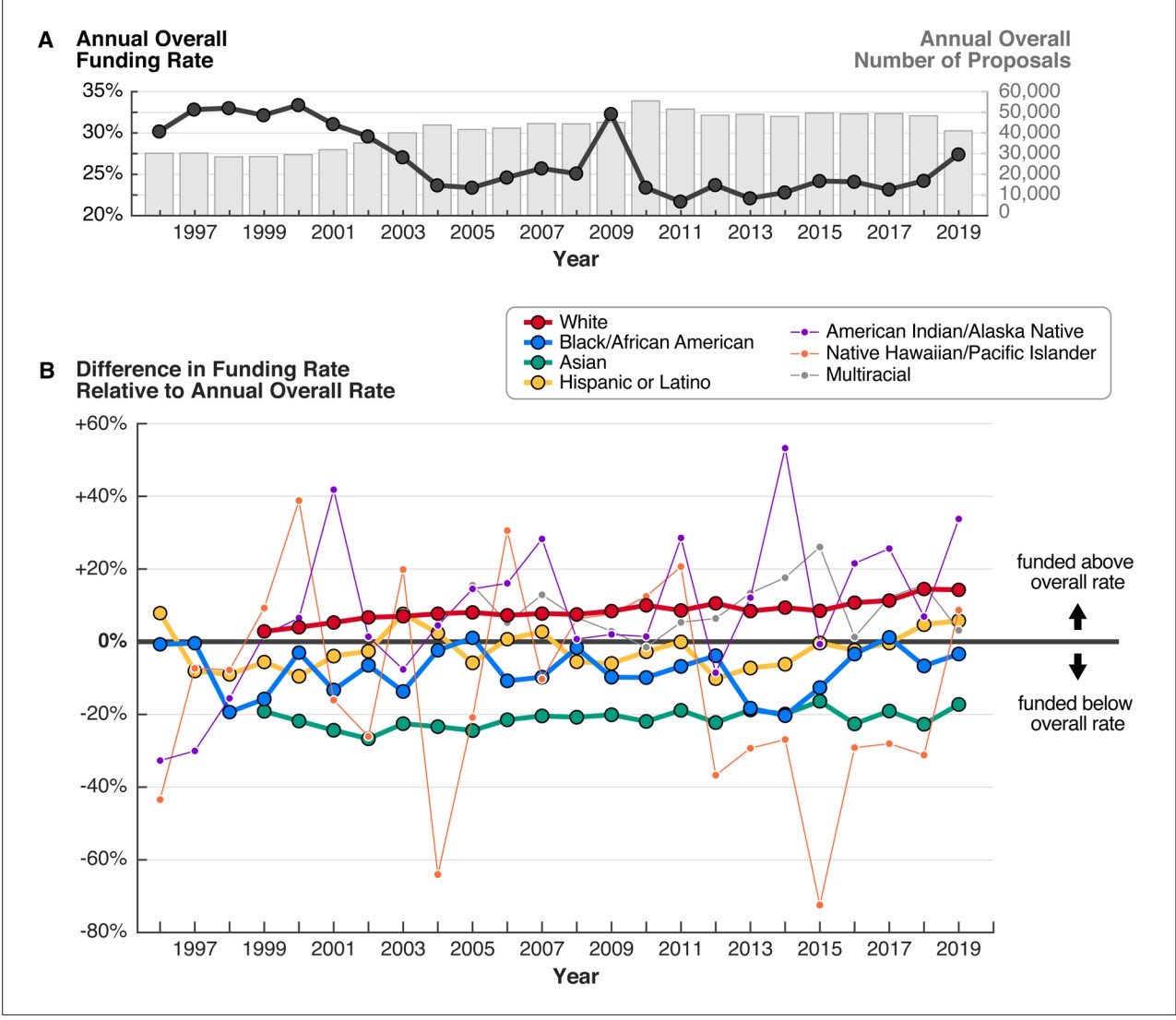

**Figure 1.** From 1999 to 2019, proposals by white PIs were consistently funded at rates above the overall average, while proposals by most other groups were funded at rates below the overall average. (**A**) Overall funding rates (black line) and total number of proposals (gray bars) have fluctuated on a yearly basis over time. (**B**) Racial disparities in funding rates have persisted for more than 20 years. Funding rates by PI race and ethnicity are normalized to the overall rate for each year. Groups represented by thinner lines submitted on average fewer than 500 proposals annually. Data for white and Asian PIs are only available starting in 1999, and for multiracial PIs starting in 2005. Source data: Data S1 in the accompanying data repository (https://doi.org/10.5061/dryad.2fqz612rt).

The online version of this article includes the following figure supplement(s) for figure 1:

**Figure supplement 1.** Jitter plot illustrating the variance of annual overall relative funding rates for all proposals by PI race and ethnicity, 1999–2019.

**Figure supplement 2.** The proportions of all proposals and all awards by PI race and ethnicity, 1996–2019.

**Figure supplement 3.** Funding rates and relative funding rates for underrepresented racial and ethnic minority PIs.

### Racial stratification in award types

Research awards are the standard mechanism through which the NSF funds PIs and institutions, comprising 71–76% of all awards from 2013 to 2019 (*Figure 3B*). The remaining 24–29% of awards consist of grants for other activities and expenses, such as exploratory or early concept work; education and training; equipment, instrumentation, conferences, and symposia; and operation costs for facilities. The funding rate for these types of proposals, categorized by NSF as "Non-Research," is generally 1.4–1.9 times higher than that for "Research" proposals (*Figure 3A*).

When we examine racial disparities in the context of Research and Non-Research proposals

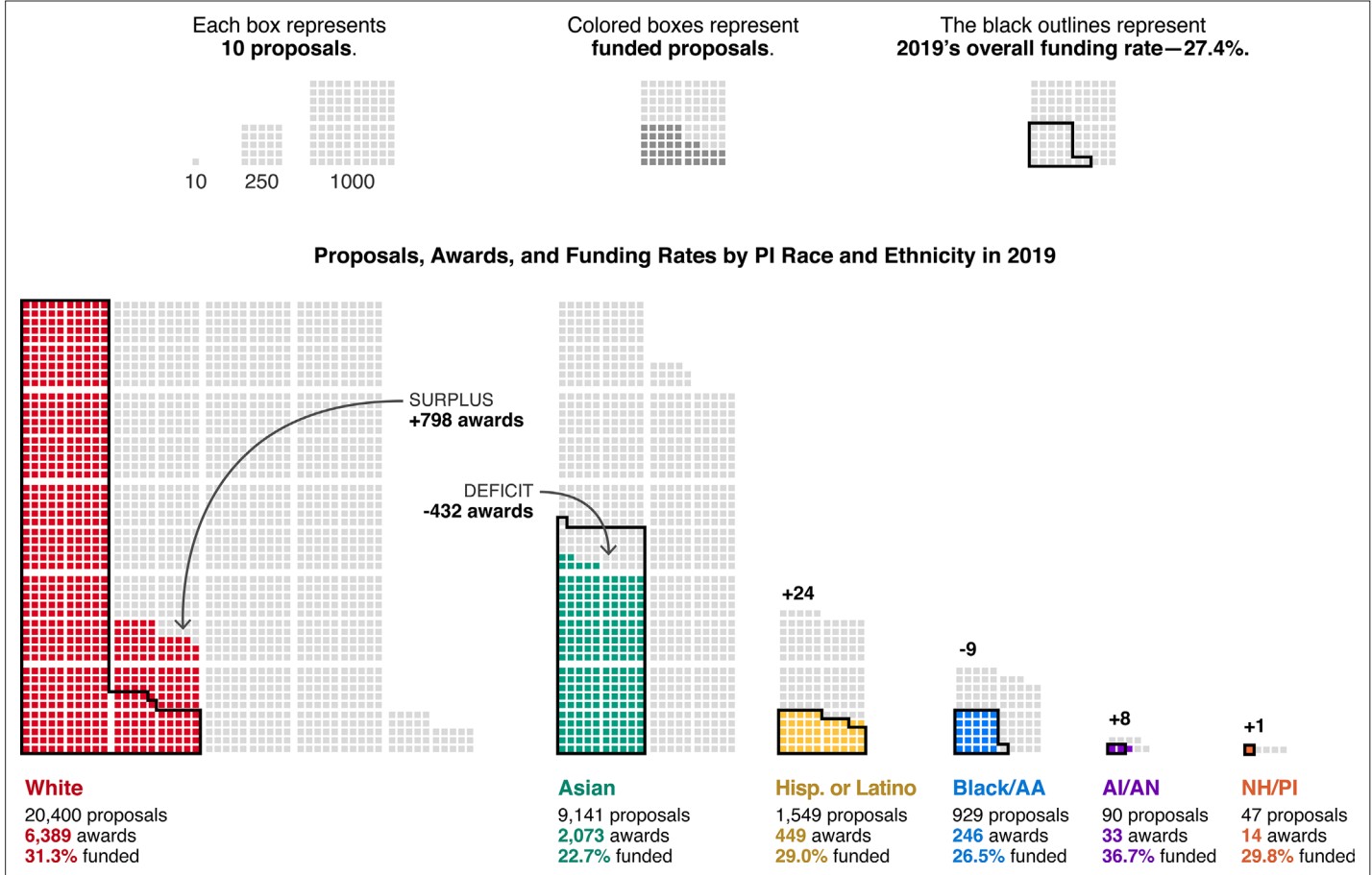

**Figure 2.** In 2019, racial disparities in funding rates corresponded to hundreds of awards in surplus to white PIs and hundreds of awards in deficit to other groups. Each box represents 10 proposals. Light gray boxes are unsuccessful proposals; colored boxes are funded proposals (awards). The black outlines represent 27.4% of the proposals submitted by each group, where 27.4% is the overall funding rate in 2019. For each group, the number of awards above (surplus) or below (deficit) this threshold is in bold. This graphic does not include proposals by multiracial PIs or PIs who did not provide their race or ethnicity. Source data: Data S1 in the accompanying data repository (https://doi.org/10.5061/dryad.2fqz612rt).

from 2013 to 2019 (the years with available data), we find that disparities for Research proposals are generally larger (*Figure 3C*). White PIs were the only group whose Research and Non-Research proposals were consistently funded above overall rates. The relative funding rates for Research and Non-Research proposals by white PIs also gradually increased, from +9.0% and +5.6% in 2013 to +14.8% and +13.2% in 2019, respectively. In addition, for most years, Research proposals by white PIs had higher relative funding rates than Non-Research proposals.

In contrast, Research proposals by PIs from nearly every other racial and ethnic group had negative relative funding rates, and were generally funded at lower rates compared to Non-Research proposals (*Figure 3C*). In particular, for Black/AA PIs, the relative funding rates for Research proposals in 2013 and 2014 were anomalously low, at –35.2% and –38.9%. These

low funding rates meant that Research proposals by white PIs were funded 1.7 and 1.8 times more than those by Black/AA PIs in these years, with relative funding rates of +9.0% and +10.6% (absolute funding rates of 21.3% and 22.6% white versus 12.6% and 12.4% Black/AA). For Asian PIs, relative funding rates for Research proposals fluctuated between –24.0% and –14.2%, for an average of –19.1% from 2013 to 2019. Whereas the relative Research proposal funding rate for Black/AA PIs gradually increased to –9.9% in 2019, the rate for Asian PIs did not. Similar or worse outcomes are observed for Research proposals by NH/PI PIs, especially in 2015, when only 1 of 23 Research proposals were awarded (4.3%).

These Research funding rate disparities contribute to a stratification in awarded activities by race: from 2013 to 2019, only 46–63% of awards to Black/AA PIs were for Research.

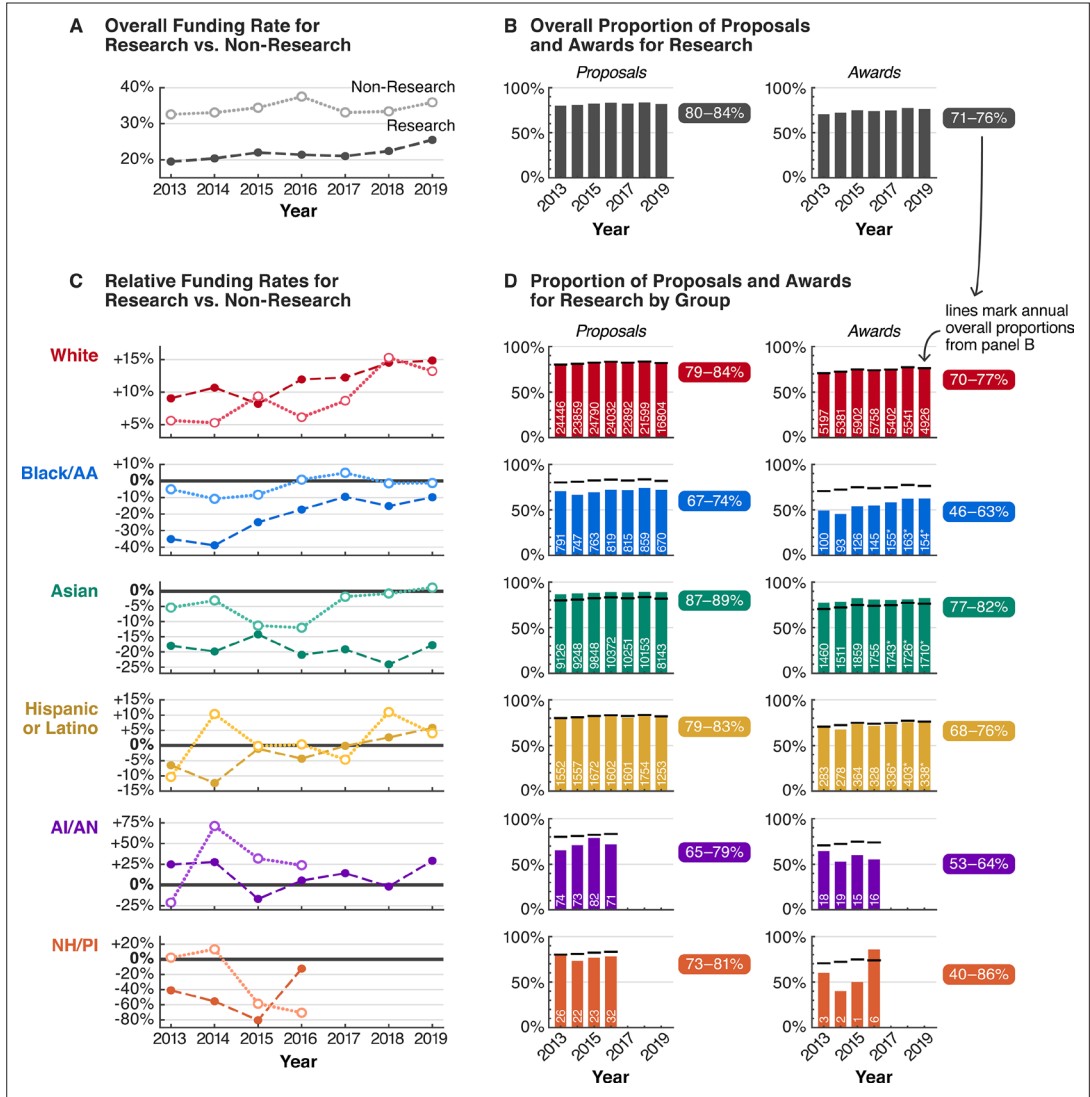

**Figure 3.** From 2013 to 2019, racial funding disparities were even greater for Research proposals, contributing to racial stratification in Research versus Non-Research activities. (**A**) Overall funding rates for Research proposals (dark dashed line) are more competitive than overall funding rates for Non-Research proposals (light dotted line). (**B**) 80–84% of all proposals and 71–76% of all awards were for Research activities. (**C**) Both Research and Non-Research proposals by white PIs were funded above overall rates. In contrast, Research proposals by PIs of most other groups were funded below overall rates, and at rates generally lower than those for Non-Research. (**D**) Only 46–63% of all awards to Black/AA PIs were for Research, far below overall proportions of awards for Research for all groups combined (black horizontal lines; panel B), contributing to a stratification of awarded activities by race. White text denotes the number of Research proposals or awards for each group per year; asterisk indicates that numbers are estimates based on available data. Non-Research data for AI/AN and NH/PI PIs were not available 2017–2019. Source data: Data S2–3 in the accompanying data repository (https://doi.org/10.5061/dryad.2fqz612rt).

The online version of this article includes the following figure supplement(s) for figure 3:

**Figure supplement 1.** Pie charts representing the percentage breakdown of all proposals from 2019 by proposal type and review mechanism.

This percentage is far below the proportion of Research awards to white PIs over the same period, 70–77% (*Figure 1D*). Although part of this stratification can be attributed to Black/AA PIs submitting proportionately more Non-Research proposals, these proposals were still consistently funded less often compared to Non-Research proposals by white PIs.

These results show that larger racial disparities in Research proposal funding rates are masked within the funding rates for all proposals. For example, in 2013, the difference in relative

funding rate for Research proposals versus all proposals for Black/AA PIs is large, –35.2% for Research compared to –18.3% for all proposals (*Figures 1B and 3C*). In addition, these results suggest that the magnitude of disparities for Asian and Black/AA PIs is more similar when considering only Research proposals: on average for the 2013–2019 period, Research proposals by white PIs had a 1.37- and 1.40-fold funding rate advantage over those by Asian and Black/AA PIs, respectively (Data S14). Thus, disaggregating funding statistics by proposal type reveals a more complete picture of these racial funding disparities and their impacts.

### Racial disparities across directorates

Likewise, examining funding rate disparities by research discipline also adds crucial context. NSF divides its research and education portfolio into seven grant-making directorates: Education and Human Resources (EHR); Social, Behavioral, and Economic Sciences (SBE); Biological Sciences (BIO); Geosciences (GEO); Computer and Information Science and Engineering (CISE); Engineering (ENG); and Mathematical and Physical Sciences (MPS). All award and decline decisions are issued through program offices and divisions specializing in distinct subfields within each directorate. Aside from differences in scientific purview, each directorate also handles varying numbers of proposals and funds them at different rates depending on their budget. As a result, overall funding rates differ between directorates, with some more competitive than others (e.g., 14.8% in EHR versus 25.5% in GEO for Research, 2012–2016). Despite these differences, from 2012 to 2016 (years with available data), after normalizing by overall directorate funding rates, we find that all directorates exhibited racial funding rate disparities and stratification patterns, albeit to varying degrees.

Most patterns for overall funding rates were also observed at the directorate level. In every directorate, proposals by white PIs were consistently funded above overall directorate funding rates, regardless of type (*Figure 4A*), and for all non-white groups, relative funding rates for Research proposals were also below those for Non-Research, with rare exceptions (e.g., GEO for Black/AA PIs). The proportion of Research awards to Black/AA PIs was also consistently below overall directorate proportions across all directorates (*Figure 4C*). Within each directorate, relative funding rates for proposals by white and Asian PIs exhibited less year-to-year variability,

owing to larger submission numbers. For other groups with fewer proposals, funding rates were more volatile, and in the case of AI/AN and NH/PIs, data were often missing (in most merit review reports, proposal or award sums fewer than 10 were omitted to protect the identities of individual investigators).

Between directorates, the magnitude of disparities and the group with the lowest funding rate varied. For Research proposals with available data, Black/AA PIs had the lowest multi-year average funding rate in CISE, EHR, ENG, SBE, and MPS, whereas Asian PIs had the lowest in BIO and GEO. Considering disparities by each year, while Research proposals by Black/AA PIs were consistently the lowest funded in CISE for every year between 2012 and 2016, the group with the lowest funding rate occasionally changed in other directorates. In comparing the magnitude of disparities for Research proposals across directorates, white PIs experienced the largest funding rate advantage over Asian PIs in BIO (1.5-fold, multi-year average) and the largest advantage over Black/AA PIs in SBE (1.7-fold, multi-year average; Data S14).

The impact of these disparities in terms of the award surpluses and deficits broadly scales with directorate size, or more specifically, by the number of proposals managed by each directorate. For example, while the +9.7% average relative Research funding rate for white PIs in MPS was not the highest of all directorates, because 21% of all Research proposals by white PIs were submitted to MPS in this period, this elevated funding rate accounted for 26% of the total Research award surplus to white PIs (117 of 441, multi-year annualized average; *Figure 4B*). However, especially high or low relative funding rates have amplifying effects. For example, although BIO received only 6% of all Research proposals by Asian PIs, because its relative Research funding rate for Asian PIs was very low (–27.8%, multi-year average), the lowest of all directorates during this period, BIO contributed 11% of the total Research award deficit to Asian PIs (34 of 306, multi-year annualized average).

These directorate funding data also reveal a paradoxical trend: relative funding rates for proposals by Black/AA PIs are *lower* for directorates with proportionally *more* proposals from Black/AA PIs (*Figure 4D*), with the exception of EHR. Although 2.4% and 2.5% of all proposals to SBE and ENG were submitted by Black/AA PIs, the multi-year average relative funding rates for proposals by Black/AA PIs in SBE and ENG were the lowest of all directorates, at –26.5% and

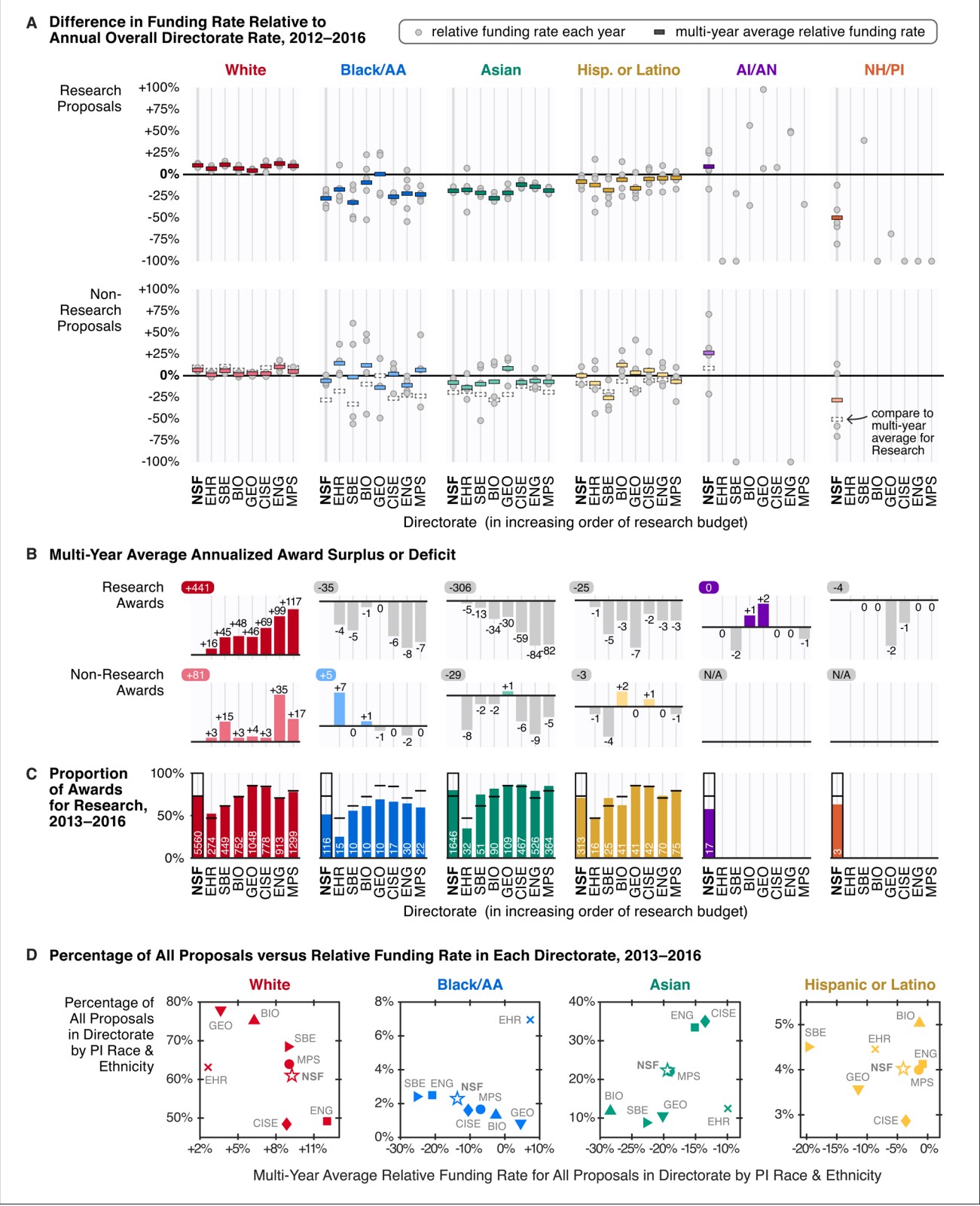

**Figure 4.** From 2012 to 2016, all disciplinary directorates exhibited racial disparities in funding rates and racial stratification in awarded activities. (**A**) Relative funding rates by directorate for Research (top) versus Non-Research (bottom) proposals by PI race and ethnicity. Gray circles mark relative funding rates for each available year; colored rectangles represent the multi-year average. To aid visual comparison, the multi-year average relative funding rate for Research proposals is superimposed on the Non-Research panel as a dotted rectangle. For Research proposals, data are available for

*Figure 4 continued on next page*

*Figure 4 continued*

at most 5 years (2012–2016); for Non-Research proposals, data are available for at most 4 years (2013–2016). (**B**) Multi-year average annualized award surplus or deficit per directorate by PI race and ethnicity, for Research (top) and Non-Research (bottom). The upper-left number in each sub-panel is the multi-year average annualized award surplus or deficit for each group for all seven directorates, excluding awards made by the Office of the Director. For AI/AN and NH/PI PIs, only data for Research awards in 2012 are shown; no directorate data for Non-Research awards are available. (**C**) Proportion of awards for Research by directorate and PI race and ethnicity, compared to overall directorate proportions (black horizontal lines), 2013–2016. White text denotes average annual number of Research awards per directorate to each group. (**D**) Percentage of all proposals submitted to each directorate by white (red), Black/AA (blue), Asian (green), and Hispanic or Latino PIs (yellow) versus the multi-year average relative funding rate for all proposals by each group, 2013–2016. Source data: Data S4 in the accompanying data repository (https://doi.org/10.5061/dryad.2fqz612rt).

The online version of this article includes the following figure supplement(s) for figure 4:

**Figure supplement 1.** Comparison of average relative funding rates for Research proposals in each directorate and the proportion of Research proposal in each directorate for white, Black/AA, Asian, and Hispanic or Latino PIs.

**Figure supplement 2.** Funding outcomes by PI race and ethnicity for all proposals in the BIO Directorate, 2012–2016.

**Figure supplement 3.** Funding outcomes by PI race and ethnicity for Research proposals in the BIO Directorate, 2012–2016.

**Figure supplement 4.** Funding outcomes by PI race and ethnicity for Non-Research proposals in the BIO Directorate, 2012–2016.

**Figure supplement 5.** Funding outcomes by PI race and ethnicity for all proposals in the CISE Directorate, 2012–2016.

**Figure supplement 6.** Funding outcomes by PI race and ethnicity for Research proposals in the CISE Directorate, 2012–2016.

**Figure supplement 7.** Funding outcomes by PI race and ethnicity for Non-Research proposals in the CISE Directorate, 2012–2016.

**Figure supplement 8.** Funding outcomes by PI race and ethnicity for all proposals in the EHR Directorate, 2012–2016.

**Figure supplement 9.** Funding outcomes by PI race and ethnicity for Research proposals in the EHR Directorate, 2012–2016.

**Figure supplement 10.** Funding outcomes by PI race and ethnicity for Non-Research proposals in the EHR Directorate, 2012–2016.

**Figure supplement 11.** Funding outcomes by PI race and ethnicity for all proposals in the ENG Directorate, 2012–2016.

**Figure supplement 12.** Funding outcomes by PI race and ethnicity for Research proposals in the ENG Directorate, 2012–2016.

**Figure supplement 13.** Funding outcomes by PI race and ethnicity for Non-Research proposals in the ENG Directorate, 2012–2016.

**Figure supplement 14.** Funding outcomes by PI race and ethnicity for all proposals in the GEO Directorate, 2012–2016.

**Figure supplement 15.** Funding outcomes by PI race and ethnicity for Research proposals in the GEO Directorate, 2012–2016.

**Figure supplement 16.** Funding outcomes by PI race and ethnicity for Non-Research proposals in the GEO Directorate, 2012–2016.

**Figure supplement 17.** Funding outcomes by PI race and ethnicity for all proposals in the MPS Directorate, 2012–2016.

**Figure supplement 18.** Funding outcomes by PI race and ethnicity for Research proposals in the MPS Directorate, 2012–2016.

**Figure supplement 19.** Funding outcomes by PI race and ethnicity for Non-Research proposals in the MPS Directorate, 2012–2016.

**Figure supplement 20.** Funding outcomes by PI race and ethnicity for all proposals in the SBE Directorate, 2012–2016.

**Figure supplement 21.** Funding outcomes by PI race and ethnicity for Research proposals in the SBE Directorate, 2012–2016.

**Figure supplement 22.** Funding outcomes by PI race and ethnicity for Non-Research proposals in the SBE Directorate, 2012–2016.

–19.6%, respectively (Data S4). The same trend is observed for Research proposals (*Figure 4— figure supplement 1*).

### Racial disparities in external review scores

To guide funding decisions, NSF program officers within each directorate oversee the vast majority of proposals through a 6-month-long external peer review process, wherein outside experts with field-specific expertise provide feedback on the merits of a proposed project. Through individual written input and/or panel deliberations, external reviewers are instructed to assess a proposal's potential to advance knowledge (intellectual merit), its potential to benefit society (broader impact), and the qualifications of the PI, collaborators, and institution (*National Science Foundation, 2021*). In addition to narrative comments, external reviewers must also give an overall rating on a scale from 'Poor' (numerically 1, "proposal has serious deficiencies") to 'Excellent' (numerically 5, "outstanding proposal in all respects"). A minimum of three pieces of external input are required for complete evaluation. While self-reported demographic data are not visible to the reviewers, PI race or ethnicity may be inferred from proposal content or personal knowledge.

Data on average review scores of externally reviewed Research proposals show that proposals by white PIs received higher scores than proposals by all other non-white groups, with scores negatively skewed, asymmetrically distributed towards higher ratings (*Figure 5A*). In 2015, the average of all average review scores for white PI proposals was 3.46 (median 3.50), compared

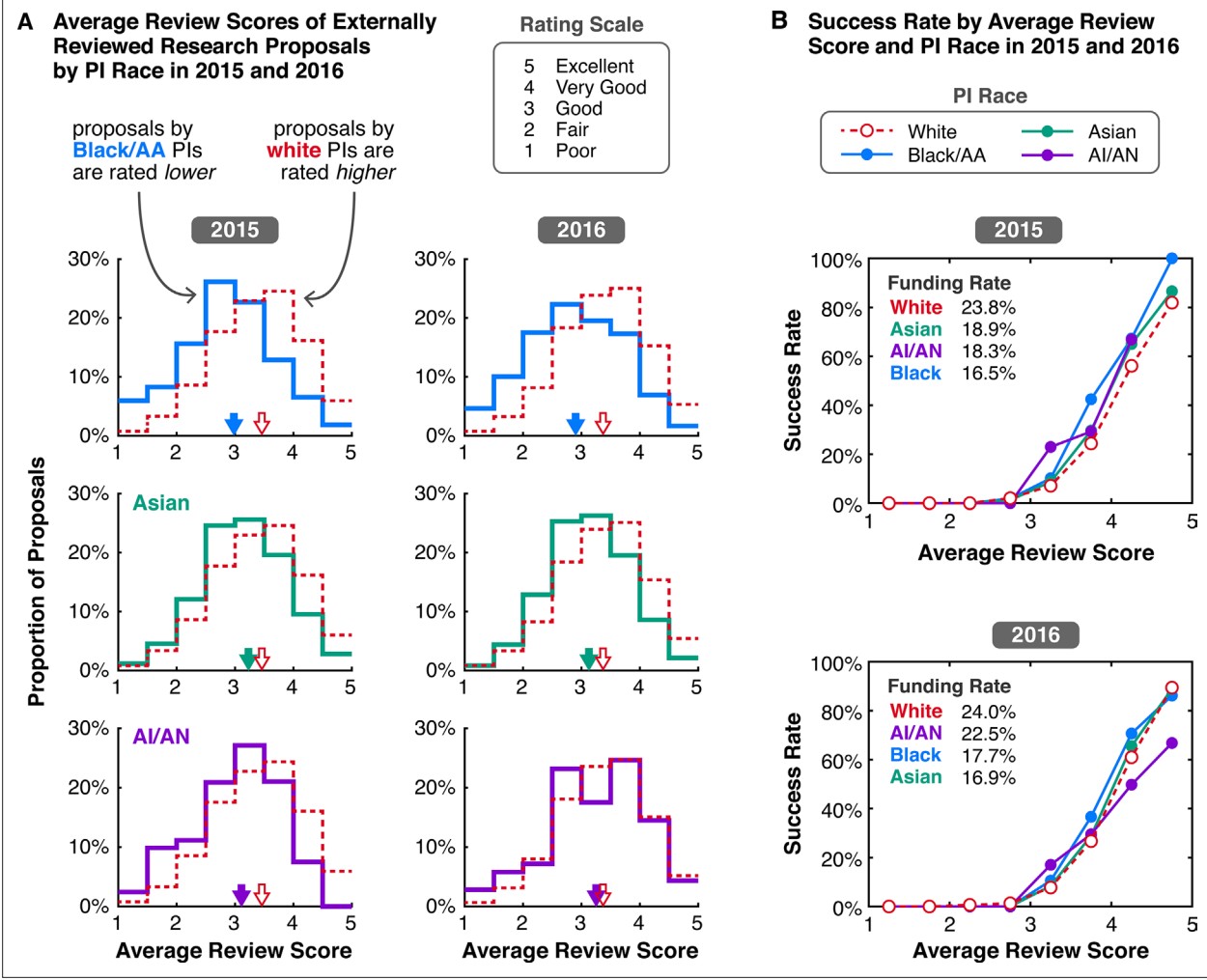

**Figure 5.** In 2015 and 2016, the distributions of average external review scores of externally reviewed Research proposals were systematically offset and skewed based on PI race. (**A**) White (red dashed), Black/AA (blue), Asian (green), and AI/AN (purple) PIs, for 2015 (left column) and 2016 (right column). Proposals are rated on a scale from 1 (Poor) to 5 (Excellent). The grand average of all average review scores for proposals by white PIs (red-outlined arrows) is higher than the grand averages of review scores for proposals by Black/AA, Asian, and AI/AN PIs (solid-colored arrows). (**B**) Funding rates of externally reviewed Research proposals by average review score and PI race. For context, the funding rate of all Research proposals by group is listed from highest to lowest in the top left corner. Data on review scores and funding rates by average score are only available for 2015 and 2016. Source data: Data S5 in the accompanying data repository (https://doi.org/10.5061/dryad.2fqz612rt).

to 2.98 for Black/AA (median 3.00), 3.10 for NH/PI (median 3.00; score distribution unavailable), 3.11 for AI/AN (median 3.33), and 3.23 for Asian PI proposals (median 3.25). Similar differences in average review scores are observed in 2016, the only other year with available data.

Accompanying information on the success rates of proposals based on average review score highlights the impact of programmatic decision-making. In 2016, although average scores for Research proposals by Black/AA PIs were lower, the relative funding rate for Research proposals by Asian PIs was worse, −20.9% for Asian PIs compared to −17.3% for Black/AA PIs (*Figure 3C*). This counterintuitive result may be

attributed to differences in success rates for proposals with comparable scores. Although proposals with higher scores are more likely to be awarded, the success rates for Research proposals by Black/AA PIs were generally higher than those for white and Asian PIs with the same score (*Figure 5B*). For Asian PIs, success rates for Research proposals are not as high, closer to the rates for white PIs. These decisions to fund proposals outside of their rank order by score reflect NSF's discretion to consider scores alongside other factors when making funding decisions, such as reviewer comments, panel discussion summaries, and a need to balance a diverse research portfolio in line with the agency's

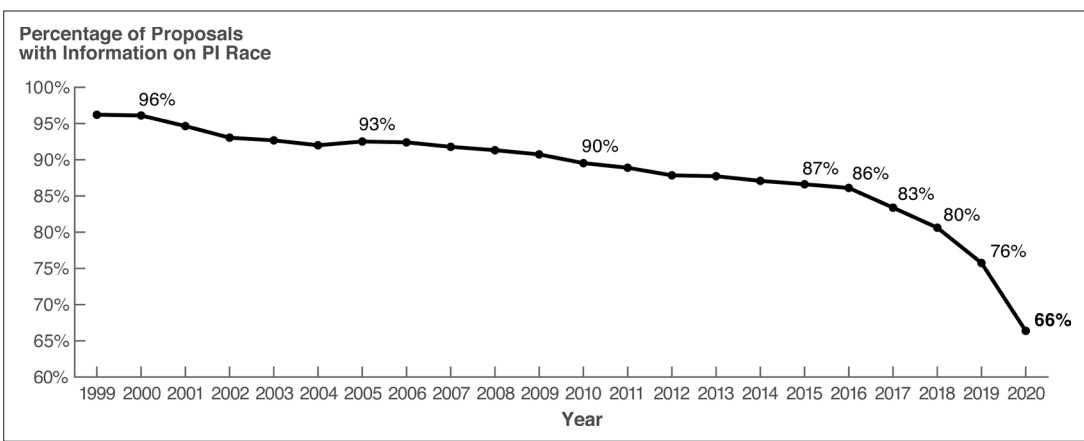

**Figure 6.** The decline in the proportion of proposals by PIs who identified their race has accelerated in recent years. Source data: Data for 1999–2019 are collated in Data S1 in the accompanying data repository (https://doi.org/10.5061/dryad.2fqz612rt); data for the year 2020 are available from the 2020 NSF report on the Merit Review Process (**National Science Board, 2021a**).

The online version of this article includes the following figure supplement(s) for figure 6:

**Figure supplement 1.** Availability of data on funding outcomes by PI race and ethnicity and other demographic characteristics in NSF merit review reports, 1990–2019.

**Figure supplement 2.** The impact of changes in the tabulation of PI race and ethnicity data on relative funding rates, 1996–2019.

**Figure supplement 3.** Proposals, awards, and relative funding rates for New PIs versus Prior PIs (1990–2019) and Early Career PIs versus Later Career PIs (2000–2019).

**Figure supplement 4.** Proposals, awards, and relative funding rates for PIs at Minority Serving Institutions.

**Figure supplement 5.** Proposals, awards, funding rates, and relative funding rates for PIs in Established Program to Stimulate Competitive Research Jurisdictions, 2001–2019.

**Figure supplement 6.** Available statistics on a per-PI basis for Research proposals and awards in three-year windows, 1995–2019.

**Figure supplement 7.** Additional trends in the non-reporting of demographic information by PIs, 1999–2019.

statutory mission and national interest (**National Science Foundation, 2021**). Although proposals by white PIs generally experience lower success rates by score compared to most other groups, the large absolute number of proposals by white PIs combined with their above average scores still resulted in relative Research funding rates of +8.2% and +12.0% for white PIs in 2015 and 2016.

### *Limitations of data*

These racial funding disparities raise many questions about their underlying causes and mechanisms, but limitations of current publicly available data reported by NSF restrict such inquiries from being robustly investigated. Since 2003, NSF has used the racial and ethnic categories and definitions set by the Office of Management and Budget in 1997, following government-wide standards for federal data collection. However, racial and ethnic categories are understood to be social constructs with no biological basis, and as

such, are complex and highly mutable over time, subject to changes in social perceptions of race and self-identification (**Clair and Denis, 2015**). Furthermore, because our data are limited to those published in annual merit review reports, which change in content and organization each year, disaggregated information on funding outcomes by directorate and award type are only available for short time intervals, limiting our ability to fully characterize long-term trends. Lastly, modifications to the way race and ethnicity information are tabulated in merit review reports also impact data consistency (see Methods; *Figure 6—figure supplements 1–2*).

The lack of publicly available data also precludes multivariate and intersectional examinations of NSF racial funding disparities alongside other factors like gender, career stage, and institution type. Although NSF data on funding rates for PIs by these aforementioned characteristics exist, this information is tabulated separately from data by PI race and ethnicity (e.g.,

*Figure 6—figure supplements 3–5*). Due to lack of data access, we are also unable to investigate the influence of other factors that likely affect funding rates, like educational background and training, prior scholarly productivity as publications, previous funding success, mentoring networks, and institutional knowledge and support. Many of these factors have been previously shown to add crucial context to the racial funding disparities at NIH (e.g., *Ginther et al., 2011*; *Ginther et al., 2018*).

Another consideration is that data on funding outcomes by PI race and ethnicity are reported only as total numbers of proposals and awards, and do not include information on the number of unique applying PIs in each group. In any given year, a PI can submit multiple proposals and likewise receive multiple awards, making per-PI race-based differences in submission or award rates indiscernible from the available data. However, this information is reported in aggregate for Research grants in three-year windows (*Figure 6—figure supplement 6*): for example, in 2017–2019, approximately 52,600 unique PIs submitted a total of 114,655 Research proposals, and of these PIs, approximately 39.4%, or 20,700, received at least one award. For every three-year window since 1995, 34–44% of all PIs who applied for at least one Research grant received at least one award. Of these funded PIs, approximately 13–16% received two awards, and 4–5% at least three.

We also observe an emerging trend in the non-reporting of demographic information: from 1999 to 2020, the proportion of proposals submitted by PIs who provided information on their race decreased from 96% to 66% (*Figure 6*). This trend is accelerating, with a 10% drop in response rate between 2019 and 2020, the largest year-to-year decrease observed in available data. This pattern coincides with similar decreasing trends in the response rate for ethnicity and gender (*Figure 6—figure supplement 7*). The cause of this phenomenon and its prevalence elsewhere is unclear, as it has not been widely reported. Regardless, these trends are concerning, as further decreases in the proportion of respondents will undermine the statistical effectiveness of reported information, impeding future efforts to track disparities.

## Discussion

### Over 20 years of racial funding disparities at NSF, NIH, and other funding bodies

Our analysis shows that for at least two decades, there has been a consistent disparity in funding rate between proposals by white PIs and those by most other racial groups. The relative funding rate for proposals by white PIs has also been increasing with time. We further show that disparities are even greater for Research awards, a result obscured within overall statistics by higher Non-Research funding rates. Differences in the allocation of awards for Research versus Non-Research activities by racial group reveal a stratification of funded activities by race. These patterns are also observed within each directorate. Identifying the underlying causes and mechanisms for these disparities requires further study, but information on average external review scores from two recent years sheds light on processes that influence outcomes.

The racial funding disparities at NSF are comparable in magnitude, persistence, and aspect to those found in other funding bodies, despite differences in internal review processes and discipline-specific norms. In some cases, these patterns have notable similarities. For example, the 1.7- to 1.8-fold advantage for NSF Research proposals by white PIs over those by Black/AA PIs in 2013 and 2014 was likewise observed for NIH "R01"-type research proposals in 2000–2006 (*Ginther et al., 2011*), 2011–2015 (*Hoppe et al., 2019*), and 2014–2016 (*Erosheva et al., 2020*), with the same 1.7- to 1.8-fold magnitude. Our finding that racial disparities persist at the directorate level is consistent with an NIH study showing that Black/AA PIs experience both overall and within-topic funding rate disadvantages compared to white PIs (*Hoppe et al., 2019*). At the National Aeronautics and Space Administration (NASA) from 2014 to 2018, proposals by white PIs were funded at rates 1.5 times higher than those by underrepresented racial and ethnic minorities (defined by NASA as AI/AN, Black/AA, NH/PI, multiracial, and Hispanic or Latino PIs), and were considered "consistently over-selected" at "above reasonable expectation" in 2015, 2016, and 2018 (*National Academies of Sciences, Engineering, and Medicine, 2022*).

Outside the US, widening gaps in funding rates and award amounts between white and ethnic minority PIs have been documented at the Natural Environment Research Council and the

Medical Research Council, the UK counterparts to NSF and NIH (*UK Research and Innovation, 2020*). Similarly, the UK-based global research philanthropy Wellcome Trust reported a 1.9-fold disparity in funding rate between white and Black applicants from 2016 to 2020 (*Wellcome Trust, 2021*). This finding contributed to the organization's public admission of "perpetuating and exacerbating systemic racism" in 2022 (*Wellcome Trust, 2022*; *Wild, 2022*), echoing an apology made by the NIH director in 2021 for "structural racism in biomedical research" (*Kaiser, 2021*). Overall, our findings add to growing evidence that racial funding disparities within major STEM funding bodies are longstanding, persistent, and widespread.

The broad consistencies in the character of racial funding disparities at NSF, NIH, and other funding bodies suggest that similar or related mechanisms may be in effect across most, if not all, science funding contexts. A decade of research on these trends at NIH since their first accounting in 2011 offers crucial insights on mechanisms that may be active at NSF. After applying multivariate regression techniques to data extracted directly from the NIH grants management database, researchers controlled for factors that might affect grant success, such as educational background, research productivity, prior funding success, and institutional prestige, and found that applications by white PIs still had a 1.5-fold advantage over those by Black/AA PIs (*Ginther et al., 2011*). Additional dynamics disadvantaging non-white PIs were found: Black/AA and Asian PIs revised and resubmitted applications more times than white PIs before getting funded, and Black/AA PIs were also less likely to revise and resubmit a new proposal after a failed attempt due to lower review scores (*Ginther et al., 2011*; *Hoppe et al., 2019*; *Ginther et al., 2016*). Similarly, in a study of proposal submission patterns of faculty at a single medical school from 2010 to 2022, Black/AA PIs submitted 40% fewer R01 proposals than white PIs on a per-PI basis (Tables 1 and 4 from *Zimmermann et al., 2022*). Another study of both gender and race found that Black/AA and Asian women investigators were less likely to receive an R01 award than white women, highlighting a "double bind" for women of color (*Ginther et al., 2016*). And although the NIH overhauled its peer review process in 2009, disambiguating evaluative standards by requiring scores on a scale of 1–9 on five distinct criteria, an examination of 2014–2016 NIH grant applications showed that reviewers rated proposals by Black/AA PIs systematically lower on all criteria

(*Erosheva et al., 2020*). This result mirrors our findings for external reviews of 2015–2016 NSF Research proposals (*Figure 5*).

Although the currently available NSF funding data do not permit similar multivariate analyses, research on NSF funding outcomes by gender shows that although women were as or more likely to be funded compared to men, women submitted fewer proposals than expected, were less likely to reapply after a failed proposal, and were more involved in education and teaching activities than men (*Rissler et al., 2020*). Our finding that disproportionately more awards to Black/AA PIs are for Non-Research endeavors is reminiscent of this lattermost finding, and of concern given the prestige differential between Research and Non-Research output like education and training.

Future work that leverages differences and similarities in review processes and funding outcomes between NSF, NIH, NASA, and other funding bodies may yield insights on causal mechanisms and offer potential solutions. For example, at face value, Asian PIs have the lowest overall proposal funding rates of all racial groups at NSF while Black/AA PIs have the lowest rates at NIH. While this observation is not false, the overall funding rate disadvantage for Asian PIs compared to white PIs is similar in magnitude at both agencies: NIH research project grant (RPG) proposals by white PIs were funded 1.3 times more than those by Asian PIs in the 2010–2021 period (*Lauer et al., 2022*), compared to 1.4 times more for NSF proposals from 2010 to 2020 (2020 data from *National Science Board, 2021a*; data for 2021 unavailable). In contrast, over the same time periods, the funding rate advantage for white PIs compared to Black/AA PIs was 1.7-fold at NIH compared to 1.2-fold at NSF. In other words, the funding disadvantage experienced by Black/AA PIs compared to white PIs is worse at NIH than at NSF. These NIH and NSF data are not wholly equivalent: NIH RPGs only include grants for research activities, and as previously shown, overall NSF proposal outcomes mask larger disparities for Research proposals. However, in the NSF BIO directorate, which has the most disciplinary overlap with the NIH, Research proposals by white PIs had a 1.5- and 1.2-fold advantage over those by Asian and Black/AA PIs in the 2012–2016 period (*Figure 4A*). The recurrence of the pattern in the BIO directorate suggests that interagency differences in overall Black/AA funding outcomes may remain even if more equivalent data were available for comparison.

One explanation for this difference may lie in the panel discussion and decision-making phases of the merit review process. At NSF, success rates based on review scores in 2015 and 2016 indicate that funding decisions partially countered the lower scores of proposals by Black/AA PIs, with a smaller effect on proposals by Asian PIs (*Figure 5B*). This effect has also been observed in other funding contexts where unequal evaluations by gender were counteracted by panels, leading to gender-equal funding outcomes (*Bol et al., 2022*; *van de Besselaar and Mom, 2020*). In contrast, at NIH, proposals by white PIs are often funded at higher rates than those by Black/AA PIs with comparable scores, a pattern that persists within research topic clusters (Table 1 and Figure S6 in *Hoppe et al., 2019*). Whether the countering effect at NSF is primarily occurring at the panel discussion stage, when NSF program officers issue an award or decline recommendation, or when division directors make a final decision is unknown with currently available information. Nevertheless, differences in the way NIH exercises their prerogative to fund proposals outside of rank order likely contributes to the discrepancy in Black/AA PI funding outcomes between NSF and NIH (*Taffe and Gilpin, 2021*).

### The need for disaggregation and expanded approaches to evaluating demographic progress

The importance of data disaggregation is not only demonstrated by the finding of larger racial disparities for Research proposals and directorate-level patterns, but also by funding rates for proposals by PIs who are underrepresented racial and ethnic minorities (URMs). At NSF, the URM category consists of Black/AA, AI/AN, NH/PI, and Hispanic or Latino (excluding non-Hispanic white and Asian), and is used to track and allocate resources to programs aimed at broadening participation in STEM. While the relative funding rate for URM PIs has improved and remained close to the overall rate for many years (average relative funding rate of –0.2% in 2015–2019; *Figure 1—figure supplement 3*), this metric masks important funding rate differences between constituent groups, aliasing the negative relative funding rates experienced by Black/AA PIs. In this way, URM aggregation diverts focus away from specific interventions that might address unique barriers to the success of Black/AA PIs (*Williams et al., 2015*; *Leslie et al., 2015*; *McGee, 2021*). URM aggregation also compounds the erasure of groups with

relatively low numbers, such as AI/AN and NH/PI PIs, hindering our ability to understand and mitigate racial disparities that affect Indigenous groups (*Peters, 2011*).

Reliance on the URM category may have also influenced NSF funding outcomes for Asian PIs, who are not considered URM yet experience the largest disparity in funding rates amongst all non-white racial groups (*Figure 1B*). The magnitude and lack of improvement in relative funding rates across the entire study period suggests an inadequacy of attention to this group, possibly influenced by the 'model minority' myth that Asians do not face academic challenges (*Poon et al., 2016*; *Kim, 1999*). Grouping Asians into a single racial category also overlooks heterogeneity in underrepresentation of certain groups not included in NSF diversity programs (e.g., Hmong, Filipino, Vietnamese, Sri Lankan, Bangladeshi). Narratives of Asian overrepresentation further ignore underrepresentation in several STEM subdisciplines, such as ecology and evolutionary biology and certain geoscience disciplines (*Nguyen et al., 2022*; *Bernard and Cooperdock, 2018*).

These results emphasize the need to expand approaches to studying, measuring, and evaluating demographic progress in STEM, as primary approaches presently used are vulnerable to the same exclusionary tendencies that such work seeks to remedy (*Metcalf et al., 2018*; *Dean-Coffey, 2018*; *Hanna et al., 2020*). The widespread use of the URM category is a cautionary example of statistical significance being prioritized at the expense of individuals from groups with less representation, leading to problematic mergers of categories that dilute multifaceted experiences into simplistic counts and proportions (*Mukherji et al., 2017*; *McCloskey and Ziliak, 2008*). Furthermore, future work on racial disparities in STEM funding must move away from deficit-oriented framings that have largely fallen out of favor in higher education research, and instead look towards structural mechanisms that affect outcomes (*Valencia, 1997*; *Kolluri and Tichavakunda, 2022*). For example, through the organizationally facilitated distribution of resources (*Ray, 2019a*), NSF's application of racial and aggregated URM categories may have exacerbated the racial disparities themselves.

Notably, these data counter the common assumption that achieving representation is sufficient to resolve issues of inequality. Two observations underscore this point: the large disparities in funding outcomes for Asian PIs (*Figures 1B, 3C and 4A*) and the inverse relationship across directorates between the proportion

of proposals by Black/AA PIs and their relative funding rates (*Figure 4D*). SBE disciplines, which include sociology, psychology, and economics, have a higher percentage of Black/AA scholars compared to traditional STEM fields (*Hur et al., 2017*), yet the SBE directorate yields the greatest white-Black/AA funding rate disparity of all directorates, with white PIs experiencing a 1.7-fold Research proposal funding rate advantage over Black/AA PIs. These outcomes parallel other observations of continued or increased gender bias in disciplines where women have become better represented (*van der Lee and Ellemers, 2015*; *Begeny et al., 2020*; *Huang et al., 2020*).

The cause of this inverse trend across directorates requires further examination, but a recent study showing author-reviewer homophily in peer review suggests that reviewer, panel, and program manager demographics may play a role (*Murray et al., 2019*). Preferences for shared characteristics may benefit PI applicants with identity markers that are overrepresented amongst evaluators, producing a feedback loop if prior funding success is a desired qualification for reviewer or panel participation (*National Institutes of Health, 2015*). We note however that some studies have shown an opposite effect, in which women reviewers and panelists exhibit a stronger bias against women applicants (*van de Besselaar and Mom, 2020*; *Broder, 1993*), indicating that author-reviewer dynamics are complex and require further study.

### Decades of cumulative advantage and disadvantage at the NSF

These results paint a stark picture of racial inequality in scientific funding, a finding that is more alarming when considering their compounding impact. At the individual level, because grant reviewers must use past achievements as indicators of a proposing investigator's qualifications, increased research productivity from previous awards contributes positively to subsequent grant-seeking pursuits. This "rich-get-richer" phenomenon or "Matthew effect," in which past success begets future success, has been widely documented in science since the 1960s (*Merton, 1968*; *Merton, 1988*; *Petersen et al., 2011*; *Way et al., 2019*). A recent study of an early career funding program found that winning applicants just above a threshold later secured twice as much funding than non-winners narrowly below the cutoff, highlighting the divergent impacts of early funding success or failure (*Bol et al., 2018*). Such effects have contributed

to rising inequality in biomedical research funding, where a decades-long continuous drop in the number of young PIs in biomedicine has coincided with an increasing concentration of NIH funding given to elite PIs and institutions (*Levitt and Levitt, 2017*; *Wahls, 2019*; *Katz and Matter, 2020*; *Lauer and Roychowdhury, 2021*; *Lauer, 2022*). Because non-white PIs must invest more time and energy than white PIs for funding access at every career stage (*Ginther et al., 2011*; *Hoppe et al., 2019*; *Erosheva et al., 2020*; *Ginther et al., 2018*; *Ginther et al., 2016*; *Wellcome Trust, 2021*; *National Academies of Sciences, Engineering, and Medicine, 2022*; *UK Research and Innovation, 2020*; *Lauer, 2021*), non-white PIs are less likely to be the beneficiaries of such additive advantages.

While additional work is required to understand the full consequences of the multi-decadal racial funding disparities at NSF, some elements of their cumulative impact are quantifiable. If we consider the annual award surplus or deficit to each group from 1999 to 2019, white PIs received between 203 and 904 awards in surplus each year (*Figure 7A*), a number that has increased with time due to a steady increase in relative funding rate (*Figure 1B*). Meanwhile, Asian PIs were consistently underfunded relative to the number of proposals submitted, with annual award deficits between 239 and 625 over the same period. For Black/AA, Hispanic or Latino, NH/PI, and AI/AN PIs, the average annual number of awards granted above or below overall rates was −20, −8, −1, and +4, respectively (*Figure 7—figure supplement 1*). Considered cumulatively, these quantities amount to thousands of funded or rejected proposals over the past two decades (*Figure 7B and C*).

In terms of grant dollars, this cumulative award disparity may represent several billions of dollars in unbalanced funding, based on average award size data from NSF financial reports from 1999 to 2019. In inflation-adjusted dollars, the average annualized award size for competitive grants increased from $138,300 in 1999 to $197,500 in 2019, while the average award duration varied between 2.5 and 3.5 years (*Figure 7—figure supplement 2*). Moreover, the median award size for Research awards is larger than the median for Non-Research awards. Given the previous observation that racially disparate funding outcomes for Research proposals are larger (*Figure 3C*) and that proportionally more awards to Black/AA, AI/AN, and NH/PI are Non-Research awards (*Figure 3D*), the long-term pecuniary disadvantages for Black/AA, NH/PI, and AI/AN PIs may

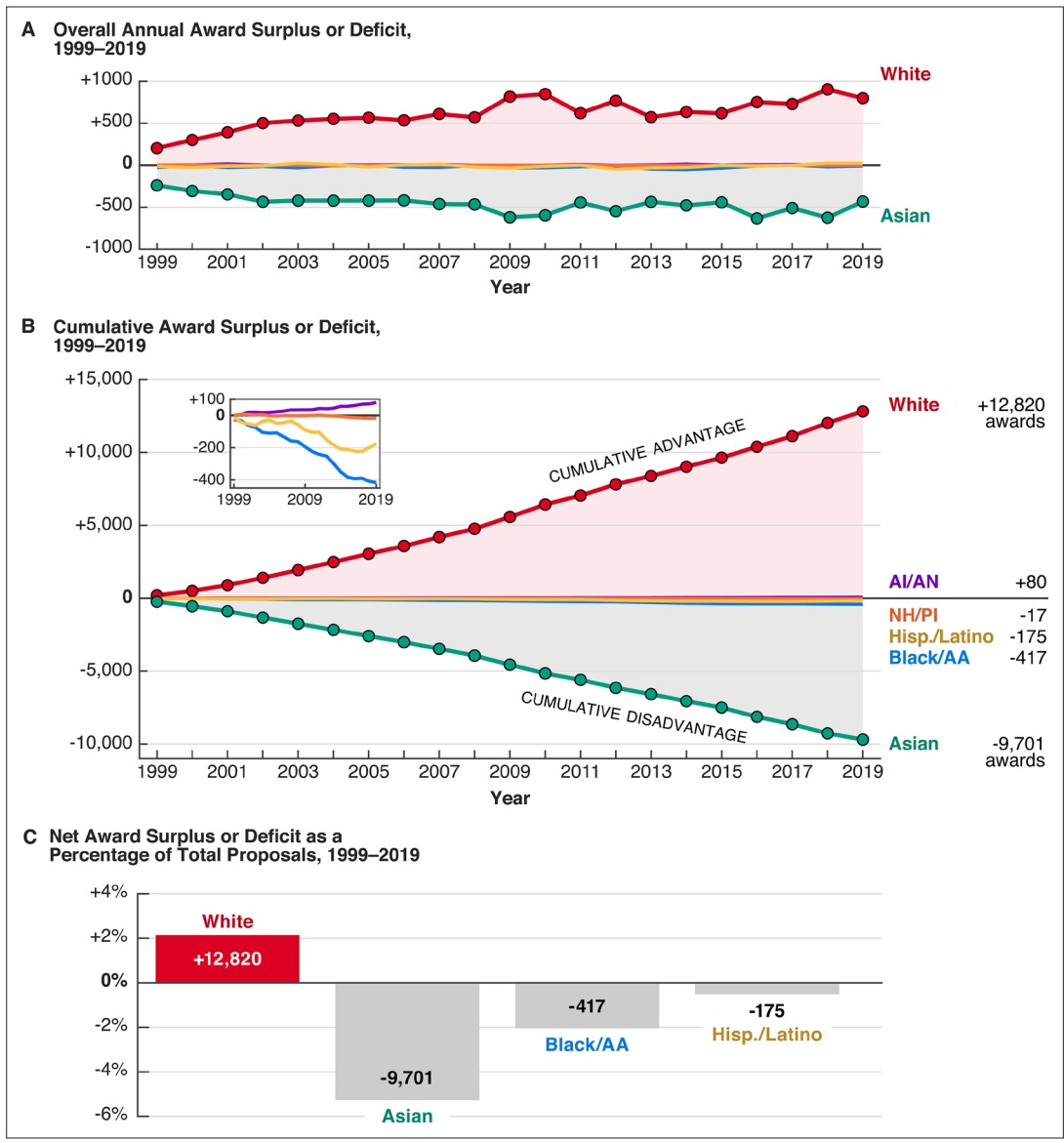

**Figure 7.** Over 20 years of racially disparate funding outcomes confer a cumulative advantage on white PIs and a cumulative disadvantage on most other groups. (**A**) Both the annual award surplus to white PIs and the annual award deficit to Asian PIs has increased over time. All other groups have annual award surpluses or deficits between –60 and +30 (*Figure 7—figure supplement 1*). (**B**) The cumulative award surplus or deficit to various groups represents thousands of awards received or not given. The small inset shows cumulative numbers for AI/AN, NH/PI, Hispanic or Latino, and Black/AA PIs. (**C**) Relative to the total number of proposals submitted by each group from 1999 to 2019, the net award surplus or deficit for white, Asian, Black/AA, and Hispanic or Latino PIs.

The online version of this article includes the following figure supplement(s) for figure 7:

**Figure supplement 1.** Overall annual and cumulative award surplus and deficit for Black/AA, AI/AN, NH/PI, and Hispanic or Latino PIs, 1999–2019.

**Figure supplement 2.** Median and average annualized award size and average award duration for all awards and Research awards, 1998–2020.

be further compounded. While these award surpluses and deficits must be considered by proposal type and program, the general picture of cumulative impacts from persistent funding rate differences is indisputable.

These trends represent just one facet of the series of interdependent systems in STEM that manifest unequal outcomes. A litany of prior work shows that while PIs of certain dominant or majority groups benefit from a system of

cumulative advantage, particularly white men at elite institutions (*Wahls, 2019*; *Katz and Matter, 2020*; *Cech, 2022*; *Sheltzer and Smith, 2014*), those of underrepresented or historically excluded groups are systematically burdened with barriers at every stage of their professional development—from placement into lower-prestige institutions as faculty (*Clauset et al., 2015*), smaller institutional start-up funds (*Sege et al., 2015*), smaller and less beneficial collaboration networks (*Ginther et al., 2018*; *Warner et al., 2016*; *Rubin and O'Connor, 2018*), disproportionate service expectations (*Jimenez et al., 2019*), lower salaries (*Cech, 2022*; *Thomson et al., 2021*), increased scrutiny and tokenization (*Settles et al., 2019*), and added stressors in suboptimal work environments (*Eagan and Garvey, 2015*), to gaps in citations, publications, promotions, and peer recognition that increase with career stage (*Ginther et al., 2018*; *Huang et al., 2020*; *Eagan and Garvey, 2015*; *Mendoza-Denton et al., 2017*; *Roksa et al., 2022*; *Hofstra et al., 2020*; *Kozlowski et al., 2022*; *Larivière et al., 2013*; *West et al., 2013*; *Bertolero et al., 2020*; *Settles et al., 2021*; *Settles et al., 2022*). Together, these barriers traumatize researchers (*McGee, 2021*), aggravate attrition (*Huang et al., 2020*; *Hofstra et al., 2020*; *Settles et al., 2022*), and impair health (*Zambrana, 2018*). The synthesis of these interlocking dynamics magnifies and perpetuates a cycle of funding disadvantage for marginalized researchers, functioning as both a cause and effect of the racial funding disparities described herein.

Given the central role that funding plays in the longevity of a researcher's career, the cumulative impact of these widespread funding inequalities has likely been paramount in shaping the racial and ethnic demographics of tenure-track and tenured faculty in STEM and academia, which have not meaningfully changed over the past decade (*Matias et al., 2021*). The metaphorical "leaky pipeline" model, which attributes the paucity of underrepresented faculty to a lack of available talent and passive attrition, fails to capture the realities of an unequal system that disproportionately supports some while diminishing or excluding others (*Berhe et al., 2022*). Although improving the diversity of the STEM workforce has long been a priority at the NSF, NIH, and other funding organizations, such goals cannot be achieved under widespread conditions that compound advantages for dominant majority groups.

## Conclusions

### Grand challenges in achieving racial equity at the NSF

As the federal steward for basic research and science workforce development, the NSF must lead efforts to achieve racial equity in STEM, modeling the change it aspires to see in other organizations and sectors. We highlight key areas that must be addressed to make funding more equitable, to the benefit of the scientific workforce and all of society. We note that these recommendations are process-oriented rather than prescriptive.

Improve data transparency and use equity metrics

Historically, NSF leads most research funding organizations in data transparency. The funding data that made this study possible is publicly available and accessible in a way that has not been emulated at most other funding bodies. NSF must continue to set an example and improve transparency by making all funding data disaggregated by race and ethnicity consistent, comprehensive, and publicly available, where possible (*de Souza Briggs and McGahey, 2022*). Privacy concerns around disaggregating and releasing data for groups with small numbers can be ameliorated by releasing data as multi-year averages or obtaining PI consent (*Taffe and Gilpin, 2021*). Additional work to understand underlying causes of funding disparities will require an intersectional approach (*Crenshaw, 1989*), investigating outcomes along multiple axes of identity and background, including but not limited to race, gender, disability, age, career stage, citizenship status, educational history, institution, and socioeconomic background (*Rissler et al., 2020*; *Williams et al., 2015*; *Leggon, 2006*; *Institute of Medicine, 2013*; *National Academics of Sciences, Engineering, and Medicine, 2019*; *CEOSE, 2020*). NSF must also expand approaches to measuring and evaluating progress towards equity (*Metcalf et al., 2018*; *Dean-Coffey, 2018*). Such research is critical for informing policies and programs aimed at addressing disparities, which risk being overly simplistic or even counterproductive without such contextualizing information.

Increase funding and accountability for equity efforts

We are encouraged by ongoing conversations that focus on improved guidance for broadening

participation in proposal review criteria and expanded programming that enhances inclusion. Past outcomes from broadening participation activities should be reported as part of grant applications. Proposals can include evidence of healthy work environments, from the establishment and tracking of equity metrics (*de Souza Briggs and McGahey, 2022*; *de Souza Briggs et al., 2022*), to improved diversity of leadership, workforce, and trainees. NSF should continue to expand opportunities for direct funding towards equity research (e.g., INCLUDES, TCUP), both to better understand disparities and their causal mechanisms, as well as to address bias inherent to how racial disparities are commonly studied and funded. These measures must be combined with intentional efforts to create equitable funding outcomes.

Eliminate the impacts of racial funding disparities

Changes in funding agency policies, practices, and resource allocation are essential to addressing disparities, as outlined by numerous calls to action by coalitions of scientists (*Stevens et al., 2021*; *No Time for Silence, 2020*; *Tilghman et al., 2021*; *Graves et al., 2022*). A decade of efforts by NIH and more recent efforts by the Wellcome Trust have demonstrated that interventions focused solely on individual actions, such as increased bias-awareness training, or specific decision points within the merit review process, like blinding peer review, are inadequate as standalone cure-all solutions (*Taffe and Gilpin, 2021*; *Wellcome Trust, 2022*; *Carter et al., 2020*; *Onyeador et al., 2021*; *Stemwedel, 2016*). The failure of these and other good-faith attempts to eradicate disparities underscores the need for multiple levels of intervention informed by a wide array of evidence-based strategies that emphasize structural change.

We urge NSF to critically reflect on these past attempts while also acting swiftly to pilot reparative measures that address these longstanding funding disparities, especially strategies that will meaningfully increase resources to diverse science and scientists. Like with public health crises and other issues of immediacy, uncertainties surrounding the exact causal mechanisms of these racial disparities should not preclude an urgent response based on what is already known (*Stemwedel, 2016*; *Kington and Ginther, 2018*). Meaningful actions can be taken while recognizing that further research and insights from intentional assessments of program efficacy will improve or change implemented strategies (*de Souza Briggs and McGahey, 2022*; *Carter et al., 2020*). Recognizing the importance of immediate actions on redressing and mitigating ongoing and future harms, in August of 2022, the Wellcome Trust announced a dedicated funding stream for researchers who are Black and people of color (*Wellcome Trust, 2022*). In the context of NSF, we note that the directorate-level data show that the number of awards needed to bridge some racial disparities is small (*Figure 4*), and that such disparities could be eliminated in a timely manner by targeted programs aimed at impacted groups.

*Examining the culture of meritocracy*

Racial funding disparities in STEM are a mirror of and magnifying glass on the ethos of meritocracy that permeates the practice of science. The use of merit review criteria to find and fund "the best ideas and the best people" is motivated by a shared understanding that the integrity of research knowledge relies on individual and collective adherence to principles of objectivity, honesty, and fairness. However, a vast body of research shows that systems designed to facilitate impartiality and merit-based rewarding can instead perpetuate the very biases they seek to prevent. For example, the issuing of a single overall rating for proposal reviews at the NSF introduces personal interpretations on the relative importance of the intellectual merit and broader impacts criteria (*Lee, 2015*; *Intemann, 2009*; *Roberts, 2009*). Additional well-documented social phenomena in evaluative STEM contexts, like "halo effects" favoring reputable scientists and institutions (*Huber et al., 2022*; *Sine et al., 2003*; *Hsiang Liao, 2017*; *Tomkins et al., 2017*) and increased bias in individuals with stronger self-perceptions of objectivity (*Begeny et al., 2020*; *Sheltzer and Smith, 2014*; *Moss-Racusin et al., 2012*; *Uhlmann and Cohen, 2007*), build on findings that environments characterized by explicit overtures of meritocracy are paradoxically more likely to produce and legitimize non-meritorious outcomes (*Moss-Racusin et al., 2012*; *Uhlmann and Cohen, 2007*; *Castilla and Benard, 2010*; *Handley et al., 2015*; *Norton et al., 2004*; *Uhlmann and Cohen, 2005*; *Apfelbaum et al., 2012*; *White-Lewis, 2020*). In this context, the racial funding disparities can be viewed as the product of a system and culture operating under an *assumed* meritocracy, rather than an aspiring one.

Other adverse impacts of a presumed meritocracy include the underfunding, under-investigation, and devaluation of ideas and topics studied by marginalized groups. The causal links between male dominance in medicine, the androcentric bias of medical knowledge, and the real-life damaging impacts on women's health have long been established (*Nielsen et al., 2017*; *Sugimoto et al., 2019*; *Koning et al., 2021*). Similarly, at the NIH, research topics that Black/AA PIs more commonly propose, such as community-oriented disease prevention, minority health, and racial health disparities, are consistently underinvested in and funded at lower rates (*Hoppe et al., 2019*; *Lauer et al., 2021*; *Taffe and Gilpin, 2021*). The devaluation of topics studied by marginalized groups has also been detected in large-scale bibliometric analyses (*Huang et al., 2020*; *Hofstra et al., 2020*; *Kozlowski et al., 2022*; *Larivière et al., 2013*) and linked to information gaps that impede fundamental inquiries into the world and universe (*Raja et al., 2022*; *Prescod-Weinstein, 2020*). These epistemic biases and inequalities in the body of scientific knowledge have cascading implications for scientific progress and the role that modern science plays in exacerbating existing societal inequities and injustices (*Settles et al., 2021*; *Settles et al., 2022*), such as race-based differences in life expectancy and health (*Bailey et al., 2021*; *Gilpin and Taffe, 2021*), disproportionate impacts of pollution and climate change (*Schell et al., 2020*; *Tessum et al., 2021*), and algorithmic racial bias in facial recognition, predictive policing, and risk-based sentencing (*Angwin et al., 2016*; *Buolamwini and Gebru, 2018*; *Benjamin, 2019*; *Hanna et al., 2020*). Although diversity-conscious efforts in STEM are often perceived as in conflict with meritocracy, or even as a threat to the core principles of science itself (*Cech, 2013*; *Posselt, 2014*), in reality, these measures intend to disrupt the forces of systemic racism that compromise the integrity of science as a public good for all.

Given that structural racism is defined as the totality of policies, processes, and social norms that interact to produce racially disparate impacts, the occurrence of racial funding disparities across STEM funding bodies serves as a warning beacon of systemic racism in science (*Rucker and Richeson, 2021*). Future efforts to understand and address these disparities must foreground the structural nature of the problem and resist conflating systemic issues with interpersonal racism (*Rucker and Richeson, 2021*; *Byrd, 2011*), like in previous responses to NIH racial funding gaps that reduced findings to a result "for" or "against" reviewer bias (*Kington and Ginther, 2018*; *Dzirasa, 2020*). While simulations of peer review show that significant differences in funding rate can result from even subtle biases (*Day, 2015*), these disparities are a reflection of the larger system of science that has conferred advantages and disadvantages in research support, publications, recognition, and influence across innumerable careers, with downstream implications for the promotion or diminishment of certain ideas. No amount of intervention focused on individual mindset change alone will undo this legacy and its influence (*Kolluri and Tichavakunda, 2022*; *Carter et al., 2020*; *Onyeador et al., 2021*; *Ray, 2019b*; *Bonilla-Silva, 2021*). Without a transformation of the historical structures that distribute power and resources for knowledge production in STEM, even in the complete absence of individual racial animus or unintended bias, these racial disparities and their harmful impacts on scientific progress will continue.

### Reimagining scientific funding

The current structures of scientific funding reflect, reinforce, and legitimate racial inequities found across society at large. Given that NSF was originally established in service of a postwar 1940s–50s America (*Wang, 1995*; *Mazuzan, 1994*; *Kevles, 1977*), in a time and place that had not yet abolished racial segregation and disenfranchisement through civil rights legislation, let alone achieved a societal shift away from attitudes favoring a strict racial hierarchy (*Rucker and Richeson, 2021*), this finding is unsurprising. The existence of widespread funding disparities both within NSF and across STEM shows that institutional racism remains readily identifiable in science and illustrates how white supremacy is maintained in contemporary contexts (*Bonilla-Silva, 2021*; *McGee, 2020*; *Bonilla-Silva, 2006*; *Bonilla-Silva, 2001*). The complete adverse consequences of these disparities on marginalized scholars, higher education, innovation in science, the scientific workforce, and society are immense and unquantifiable, but no less real. NSF and STEM at large must reckon with its own historical injustices to meaningfully challenge the status quo. To perpetuate processes that privilege whiteness is to accept as collateral damage the transgenerational loss and devaluation of contributions from marginalized groups.

Many take progress for granted and believe that issues of discrimination, bias, and inequality

will subside naturally with time (*Kraus et al., 2017*; *Kraus et al., 2019*). Yet progress towards an equitable future is not linear nor inevitable— the continuity of racial disparities across STEM funding contexts belies claims to the contrary (*Seamster and Ray, 2018*). To manifest change, NSF must lead in eliminating racial funding disparities in science with intentionality, vigilance, and a commitment to concrete and sustained action (*Richeson, 2020*). At the same time, the scientific community must engage in a full-scale re-evaluation of scientific practice and culture. Shifting the present scientific paradigm to be centered on equity will require individual, collective, and institutional commitments to elevating justice, respect, and community as core operating principles in science (*Graves et al., 2022*; *Schell et al., 2020*). Such transformation may also be advanced by an expansion beyond mainstream structures, institutions, methodologies, and ways of knowing to support and conduct scientific research (*Liboiron, 2021*; *Tuck and Guishard, 2013*). Only then can science meet the unprecedented challenges of the present and future.

## Methods

### Data sources

All data on NSF funding outcomes were extracted from annual reports on the NSF proposal review system, which are publicly available online (accessed from https://www.nsf.gov/nsb/publications/pubmeritreview.jsp). By mandate, these documents are submitted by the Director of the NSF to the National Science Board, the independent governing body of the NSF, and commonly contain information on the funding outcomes— number of proposals considered versus number of awards given—of PIs by various demographic categories like gender, race, disability, and career stage.

The NSF is authorized to collect demographic information under the NSF Act of 1950, as amended 42 U.S.C. §1861, et seq. According to information on the NSF FastLane website, this demographic data allows NSF to "gauge whether [their] programs and other opportunities in science and technology are fairly reaching and benefiting everyone regardless of demographic category; to ensure that those in underrepresented groups have the same knowledge of and access to programs, meetings, vacancies, and other educational opportunities as everyone else." The NSF collects demographic data at the time of proposal submission, when individual PIs

may opt to self-identify their racial and ethnic identity, as well as their gender, citizenship status, and disability. This information is privately held and not made available to external reviewers.

Below are the racial categories used by the NSF and their definitions (NSF FastLane, accessed June 2021), which represent categories defined by the US Office of Management and Budget (OMB) in 1997 as the minimum required response options for race and ethnicity questions in federal data collection (*US Office of Management and Budget, 1997*):

- *American Indian or Alaska Native:* A person having origins in any of the original peoples of North and South America (including Central America), and who maintains tribal affiliation or community attachment.
- *Asian:* A person having origins in any of the original peoples of the Far East, Southeast Asia, or the Indian subcontinent including, for example, Cambodia, China, India, Japan, Korea, Malaysia, Pakistan, the Philippine Islands, Thailand, and Vietnam.
- *Black or African American:* A person having origins in any of the black racial groups of Africa.
- *Native Hawaiian or Other Pacific Islander:* A person having origins in any of the original peoples of Hawaii, Guam, Samoa, or other Pacific Islands.
- *White:* A person having origins in any of the original peoples of Europe, the Middle East, or North Africa.

The NSF also defines one ethnic category:

- *Hispanic/Latino:* A person of Mexican, Puerto Rican, Cuban, South or Central American, or other Spanish culture or origin, regardless of race.

We use data from all reports that were accessible online in 2021, which includes reports from fiscal years 1996–2019. Because these reports often contain information on funding outcomes from preceding years (3–10 years, varying by report), these documents collectively provide data for various PI demographic categories from 1990 to 2019. This information is generally reported in data tables containing the number of proposals from and awards to a group.

However, because the content and organization of these reports has evolved over time, the earliest available data for certain demographic information varies. *Figure 6—figure supplement 1* graphically summarizes the demographic information available from the 1996–2019 merit review reports and their original format — a table, figure, or a description in the report text. More detailed

information on the specific reports providing specific data are described in the documentation included with the online data repository accompanying this work. For example, the NSF has reported on funding outcomes for proposals by URM PIs since the 1996 report, making 1990 the earliest year for which we have information on funding rates for URM PIs. However, funding outcomes for Black/AA, Hispanic or Latino, AI/AN, and NH/PI PIs began with the 2003 report, making 1996 the earliest year with data for these groups. Likewise, 1999 is the earliest year for which funding outcomes for white and Asian PIs are available because reporting for these categories began only in the 2007 report. It is not known to the authors whether these differences reflect changes in survey options or report content; however, federal agencies were required to make all existing demographic data consistent with 1997 OMB race and ethnicity classification standards by January 1, 2003 (*US Office of Management and Budget, 1997*).

Similarly, changes in report content and organization affect the continuity, length, and original format of other funding-related data. For example, data on funding outcomes for Research proposals by Directorate disaggregated by PI race and ethnicity are only available for the years 2012–2016. Data on average external review scores of Research proposals disaggregated by PI race are only available for the years 2015 and 2016, and are primarily shown only as a frequency distribution in figure format. We used WebPlotDigitizer (*Rohatgi, 2021*) to digitally extract the underlying numerical data in these figures. We note that although the 2015 report contained a table that reported the mean and median average review scores of Research proposals by PI race, the 2016 report did not contain a similar table.

Estimates of cumulative disparities from award surpluses and deficits in terms of total grant dollars come from data on average annualized award sizes and average award durations reported in NSF financial accountability reports, which are also publicly available online (accessed from https://www.nsf.gov/about/history/annual-reports.jsp) and are shown in *Figure 7—figure supplement 2*.

Some data on the demographic composition of reviewers are available but are limited in scope and completeness. As for PIs submitting proposals, the self-reporting of demographic information by reviewers is voluntary. According to the merit review reports, the proportion of reviewers reporting demographic information

increased from 9% in 2002 to 37.5% in 2015. Limited information about the demographics of reviewers is available only for the 2009–2015 period. Due to the incompleteness of these data, we do not examine reviewer demographics in our analysis.

### Data tabulation changes for Hispanic or Latino PIs in NSF merit review reports

Two major changes in the reporting of funding outcomes by PI race and ethnicity occurred in the 2012 merit review report (*Figure 6—figure supplement 2*). In one change, PIs who identified as Hispanic or Latino were included in both their selected racial group(s) and within the Hispanic or Latino ethnicity category. Prior to 2012, PIs who identified as Hispanic or Latino were only counted as Hispanic or Latino in merit review reports, and not included in any other race category, regardless of their selection for race. Because each report includes data from previous years, the 2012 report includes data from 2005 to 2012 with this change retroactively applied. Although not explicitly stated, this change also indicates that in reports released prior to 2012, all funding outcomes reported by PI race are for PIs who are also Non-Hispanic (e.g., white, Non-Hispanic; Asian, Non-Hispanic).

We note that although NSF describes the change in counting of Hispanic or Latino respondents as described above, there is still a mismatch between the two datasets in the number of reported proposals, awards, and funding rates for proposals by Hispanic or Latino PIs during the period of overlap 2005–2012. Given their description of the change, we would expect no discrepancies; the cause for this mismatch is unknown to the authors.

### Data tabulation changes for multiracial PIs in NSF merit review reports

In the second major change to demographic reporting in the 2012 merit review report, the NSF reported funding outcomes for PIs who selected two or more races in a separate multiracial category. PIs counted in the multiracial category are not included in any other category (i.e., an individual who selects both "white" and "Asian" for their race is placed in the multiracial category, rather than double-counted in both "white" and "Asian"). Because the 2012 report includes data on previous years, funding outcomes for multiracial PIs are available from 2005 to 2019. According to text in the 2012 report, in all reports released prior to 2012, "except for those who

were Hispanic or Latino, individuals who identified multiple races were not included in [data tables]."

### Treatment of proposals by multiple PIs in merit review reports

When PIs respond to a solicitation for proposals at NSF, PIs may submit a collaborative proposal with multiple PIs or a non-collaborative proposal as a single PI (*National Science Foundation, 2021*). In the case of collaborative proposals with multiple PIs, these may be submitted in one of two ways: as a single proposal or as multiple proposals submitted simultaneously from different organizations.

For a submission of a collaborative proposal via the single proposal method, a single PI assumes primary responsibility for the administration of the award and communication with NSF. Other involved PIs from other organizations are designated as co-PIs. If an award is given, only a single award is made. In this case, the demographic information of the PI with primary administrative responsibility is associated with the single proposal and its funding outcome.

For a submission of a collaborative proposal via simultaneous submissions from multiple organizations, separate proposal submissions are made by each PI, and the project title must begin with the words, "Collaborative Research." If the collaborative proposal is funded, each organization receives and is responsible for a separate award. In this situation, the demographic information of the PI with primary administrative responsibility for each proposal is attached to the proposal and its funding outcome. For example, if a collaborative proposal submission involves three proposals from three different PIs at different institutions, and this collaborative proposal is funded, then three proposals and three awards with the demographic information of three different PIs are included in the data tables found in the merit review reports.

Although determining the impact of collaborative research on racial funding rate disparities is of interest, currently available data do not allow for a differentiation between proposals submitted as collaborative work with multiple PIs or proposals by single PIs.

### Categories of proposals and levels of aggregation by organizational level

All proposals for which an "award" or "decline" decision has been made are sometimes categorized or referred to as "competitive" proposals in merit review reports. This category of proposals includes standard research and education proposals; conference, equipment, infrastructure, travel, and research coordination network proposals; proposals for exploratory research or in rapid response to issues of severe urgency; and other related categories. This category of proposals does not include applications for the NSF Graduate Research Fellowship, preliminary proposals, contracts, continuing grant increments, intergovernmental personnel act agreements, and other similar categories. This category also does not include proposals which were withdrawn by PIs or returned without review by program officers for not meeting certain requirements (e.g., ineligible proposals or PIs; incompliance with solicitation requirements or other policies and procedures). According to the Fiscal Year 2020 NSF Merit Review Report, typically, 1–3% of submitted proposals are returned without review.

For simplicity and ease of communication, in this paper, we have dropped the word "competitive" when we refer to all proposals for which an award or decline decision is recorded in NSF merit review reports. However, in order to maintain consistency with the original source data, the "competitive" descriptor may be retained in descriptions of collated data made available in the data repository accompanying this work.

All proposals for which NSF made an award or decline decision can be disaggregated into two types: *Research proposals* and *Non-Research proposals* (*Figure 3—figure supplement 1*). *Research proposals* comprise the majority of proposals and are the typical mechanism through which NSF funds PIs and their requests for support of research endeavors. *Non-Research proposals* are classified by NSF as proposals which are not Research proposals (i.e., total number of proposals = number of Research proposals + number of Non-Research proposals). Non-Research proposals include requests for support for education and training; operation costs for facilities; and equipment, instrumentation, conferences, and symposia. Non-Research proposals also include those submitted to the Small Business Innovation Research program. The number of proposals, awards, and funding rates for Non-Research proposals are not commonly reported, with some exceptions. In the majority of cases, for Non-Research proposals, we calculate these statistics by subtracting totals for proposals and awards for Research from the summed statistics for all proposals and awards.

Data on funding outcomes for all proposals disaggregated by PI race and ethnicity are available from 1996 to 2019 (Data S1). For all Research and Non-Research proposals disaggregated by PI race and ethnicity, data is available 2013–2019 (Data S2 and S3). For data disaggregated by directorate, funding outcomes for Research proposals are available 2012–2016, and for all proposals and Non-Research proposals, 2013–2016. Although we discuss results only for the seven disciplinary directorates, data are also available for other programs under the Office of the Director, such as the Office of Integrative Activities (Data S4).

### Data compilation and analysis

As described earlier, changes to data tabulation influence the internal consistency of some data. Two continuous and internally self-consistent datasets can be extracted from the merit review reports: a dataset for 1996–2012, in which data by race pertain to Non-Hispanic individuals only, and 2005–2019, in which data by race pertain to individuals regardless of ethnicity. *Figure 6—figure supplement 2* compares the differences between these datasets in terms of relative funding rates for proposals. For the period 2005–2012 in which there is overlap, the difference in relative funding rates between the two datasets is smaller for groups with greater numbers of proposals. To create the dataset on relative funding rates for all proposals from 1996 to 2019 (*Figure 1*), these two datasets are combined as depicted in *Figure 6—figure supplement 2*, in which data by PI race from 2005 to 2019 are regardless of ethnicity (Hispanic, Non-Hispanic, or Unknown) and data by PI race from 1996 to 2004 are only for Non-Hispanic individuals.

Funding rates for each racial and ethnic group or other categories are calculated by dividing the number of awards by the number of proposals:

$$\text{Funding Rate}_{Category} = \frac{\text{Awards}_{Category}}{\text{Proposals}_{Category}}$$

Relative funding rates for each racial or ethnic group are calculated by subtracting the overall funding rate from each group's funding rate and dividing the difference by the overall funding rates. Depending on the proposal type and organizational level being compared, the overall funding rate may be that for all proposals across NSF, all Research proposals, Research proposals in a specific directorate, *et cetera*:

$$\text{Relative Funding Rate}_{Category} = \frac{\text{Funding Rate}_{Category} - \text{Funding Rate}_{Overall}}{\text{Funding Rate}_{Overall}}$$

We also express the impact of funding rate differences by calculating the "award surplus" or "award deficit" for each group. This quantity represents the number of awards received above the overall funding rate (award surplus) by a group or the number of additional awards required for a group to be funded at the overall rate (award deficit). We calculate the award surplus or deficit by subtracting the overall funding rate from each group's funding rate and multiplying this difference by the number of proposals by each group:

$$\text{Award Surplus or Deficit}_{Category} =$$
$$\text{Proposals}_{Category} \times$$
$$(\text{Funding Rate}_{Category} - \text{Funding Rate}_{Overall})$$

A negative number indicates that the group has an award deficit whereas a positive number indicates that the group has an award surplus (*Figure 2*). Award surpluses or deficits can be calculated for specific proposal types and at different organizational levels: all proposals at NSF (e.g., *Figures 2 and 7*), for all Research proposals, Research and Non-Research proposals in a specific directorate (e.g., *Figure 4*), *et cetera*.

As previously discussed, the examination of funding rate disparities by Research proposals and by directorate illustrates the necessity of data disaggregation, as several patterns are hidden within overall funding statistics for all proposals NSF-wide. Similarly, the same is true for award surpluses and deficits. Because each group has different submission patterns in terms of the proportion of Research versus Non-Research proposals and their distribution across directorates, total award surpluses and deficits for each group that are calculated at more granular levels will differ from those calculated at broader levels. For example, for the 2019 fiscal year, if we calculate the total award surplus or deficit for white, Asian, and Black/AA PIs by separately calculating the award surplus or deficit for Research and Non-Research proposals and then summing these values, we arrive at totals of +807 for white (+637 Research, +171 Non-Research; note rounded values), –364 for Asian (–369 Research, +4 Non-Research), and –18 for Black/AA PIs (–17 Research, –1 Non-Research). These values differ from those calculated using overall funding rates

for all proposals: +798 for white, −432 for Asian, and −9 for Black/AA PIs (*Figure 2*). As a similar exercise, for the 2015 fiscal year, if we calculate the total Research award surplus for white PIs by summing the award surplus from each directorate and office (Data S4), we arrive at a total Research award surplus of +393 compared to +447 if calculated based on overall funding rates for all Research proposals. Award surpluses and deficits should be calculated at more granular organizational levels (i.e., divisions and programs) and proposal types (e.g., CAREER grants), but such data are not currently available.

### Limitations of data for PIs from racial or ethnic groups with low submission numbers

For PIs from groups with high overall submission numbers, the changes to data tabulation had a minimal to negligible overall effect on the funding rates calculated from data pre- and post-reporting scheme change. However, due to the low sample proposal numbers from AI/AN and NH/PI PIs, these data are relatively more affected by minimal changes, which include annual fluctuations in the total number of proposals awarded to these groups as well as differences in data reporting (*Figure 6—figure supplement 2*). These data are also affected by privacy and identity protection concerns in NSF reporting; information on NH/PI and AI/AN funding outcomes are often not reported when numbers are fewer than 10.

We note also that data on proposal submissions or awards for non-white Hispanic PIs is also often not reported. The groups for which data are routinely not reported — specifically, AI/AN and NH/PI — contributes to a lack of sufficient data to calculate funding rate by funded activity type (Non-Research vs. Research awards) for AI/AN and NH/PI PIs by directorate and from 2017 to 2019 overall.

### Acknowledgements
We thank the following individuals for providing feedback on early drafts of this manuscript: Krystle Palma Cobian, Katherine S Cho, Maura Dykstra, Kimberly V Lau, Michael Manga, Gabriela Serrato Marks, Rohini Shivamoggi, Frederik J Simons, and Lisa White. We thank the following individuals for feedback which led to improvements to the manuscript: Michael A Taffe and Gary McDowell for public comments and peer reviews on our preprint; the three reviewers for eLife; and an eLife editor for their careful and thorough editorial handling. CYC and SSK also thank Cin-Ty Lee for early conversations about data availability that precipitated this work. SSK thanks Breylan Martin and Jesus Nazario for thoughtful discussions on racial identity formation.

**Christine Yifeng Chen** is in the Chemical and Isotopic Signatures Group, Division of Nuclear and Chemical Sciences, Lawrence Livermore National Laboratory, Livermore, and the Center for Diverse Leadership in Science, UCLA, Los Angeles, United States
cychen@llnl.gov
http://orcid.org/0000-0002-9580-6925

**Sara S Kahanamoku** is in the Department of Integrative Biology and Museum of Paleontology, University of California, Berkeley, United States
sara.kahanamoku@berkeley.edu
http://orcid.org/0000-0003-4157-4791

**Aradhna Tripati** is in the Center for Diverse Leadership in Science, the Department of Earth, Planetary, and Space Sciences, the Department of Atmospheric and Oceanic Sciences, the Institute of the Environment and Sustainability, and the American Indian Studies Center, UCLA, Los Angeles, United States, and the Department of Earth Sciences, University of Bristol, United Kingdom
atripati@g.ucla.edu
http://orcid.org/0000-0002-1695-1754

**Rosanna A Alegado** is in the Department of Oceanography and Sea Grant College Program, and the Daniel K Inouye Center for Microbial Oceanography: Research and Education, University of Hawaiʻi at Mānoa, Honolulu, United States
rosie.alegado@hawaii.edu
http://orcid.org/0000-0003-3615-5613

**Vernon R Morris** is in the School of Mathematical and Natural Sciences, New College of Interdisciplinary Arts and Sciences, Arizona State University, Phoenix, United States
vernon.morris@asu.edu
http://orcid.org/0000-0002-6828-4017

**Karen Andrade** is in the Center for Diverse Leadership in Science, UCLA, Los Angeles, United States

**Justin Hosbey** is in the Department of City and Regional Planning, College of Environmental Design, University of California, Berkeley, United States
http://orcid.org/0000-0003-0947-4002

*Author contributions:* Christine Yifeng Chen, Conceptualization, Data curation, Software, Formal analysis, Supervision, Investigation, Visualization, Methodology, Writing – original draft, Project administration, Writing – review and editing; Sara S Kahanamoku, Conceptualization, Data curation, Investigation, Methodology, Writing – original draft, Writing – review and editing; Aradhna Tripati, Conceptualization, Writing – original draft, Writing – review and editing; Rosanna A Alegado, Conceptualization, Writing – original draft, Writing – review and editing; Vernon R Morris, Conceptualization, Writing – review and editing; Karen

Andrade, Conceptualization, Writing – original draft; Justin Hosbey, Conceptualization, Writing – review and editing

*Competing interests:* Christine Yifeng Chen: Has previously applied for, received, and benefitted from funding through grants and fellowships awarded by the NSF; has previously served as an ad hoc reviewer for NSF proposal review. Sara S Kahanamoku: Has previously applied for, received, and is currently funded through grants and a fellowship awarded by the NSF. Aradhna Tripati: Has previously applied for, received, and/or is currently funded by grants from NSF; has also previously served as ad hoc and/or panel reviewers at NSF. Rosanna A Alegado: Has previously applied for, received, and/or is currently funded by grants from NSF; has previously served as ad hoc and/or panel reviewers at NSF. Vernon R Morris: Has previously applied for, received, and/or is currently funded by grants from NSF; has previously served as ad hoc and/or panel reviewers at NSF. VRM currently serves on the Committee of Equal Opportunities in Science and Engineering to NSF. Karen Andrade: Has previously applied for and/or received grants from NSF. Justin Hosbey: Has previously applied for, received, and/or is currently funded by grants from NSF; has previously served as ad hoc and/or panel reviewers at NSF.

## Funding

| Funder | Grant reference number | Author |
| --- | --- | --- |
| National Science Foundation | DGE 2146752 | Sara S Kahanamoku |
| David and Lucile Packard Foundation | | Aradhna Tripati |
| Alfred P. Sloan Foundation | | Aradhna Tripati |
| Lawrence Livermore National Laboratory | DE-AC52-07NA27344 | Christine Yifeng Chen |

The funders had no role in study design, data collection and interpretation, or the decision to submit the work for publication.

### Decision letter and Author response

Decision letter https://doi.org/10.7554/eLife.83071.sa1
Author response https://doi.org/10.7554/eLife.83071.sa2

# Additional files

## Supplementary files

• MDAR checklist

## Data availability

All data examined in this study are publicly available in annual merit review reports published by NSF and made accessible online (https://www.nsf.gov/nsb/publications/pubmeritreview.jsp). An accompanying Dryad data repository contains collated data gathered from information contained in annual merit review reports (https://doi.org/10.5061/dryad.2fqz612rt).

The following dataset was generated:

| Author(s) | Year | Dataset URL | Database and Identifier |
| --- | --- | --- | --- |
| Chen C, Kahanamoku S, Tripati A, Alegado R, Morris V, Andrade K, Hosbey J | 2022 | http://dx.doi.org/10.5061/dryad.2fqz612rt | Dryad Digital Repository, 10.5061/dryad.2fqz612rt |

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

## Appendix 1

### The NSF merit review process
The merit review process begins upon receipt of a proposal by the NSF. Program offices operating under divisions situated within directorates issue a call for proposals for a specific subject area, and may solicit proposals on a rolling basis or by a certain deadline (*Hand, 2016*). Once a proposal is received, program officers from the appropriate program manage and oversee proposals through a six-months-long review process. NSF returns without review proposals that fail to separately address the two merit criteria, as well as ineligible proposals (duplicates of existing awards; awards that do not appropriately respond to the funding opportunity; awards that do not comply with solicitation requirements or proposal award policies and procedures; *et cetera*). Typically between 1–3% of proposals are returned without review (*National Science Board, 2021a*). Proposals returned in this way or voluntarily withdrawn by the PI are not included in the funding statistics described in the merit review reports, and by extension, are not examined in this study.

Program officers execute their responsibilities through the following actions: determining the appropriate level of merit review (Internal or External; see below); selecting ad hoc reviewers and/or panel members for review; ensuring that there are no conflicts of interest; synthesizing the comments of the reviewers and/or panel; and recommending action to either award or decline the proposal following scientific, technical, or programmatic review and consideration of additional appropriate factors (*National Science Board, 2021a*).

### Externally reviewed proposals
Approximately 95% of NSF proposals are evaluated by external reviewers (*Figure 3—figure supplement 1*). Reviewers are selected to provide program officers with the information needed to make a recommendation in accordance with merit review criteria. Ideally, reviewers should have special and broad generalized knowledge of the science or engineering subfields involved in the proposal; broad knowledge of "the infrastructure of the science and engineering enterprise, and its educational activities, to evaluate contributions to societal goals, scientific and engineering personnel, and distribution of resources to organizations and geographical areas;" and contribute to "diverse representation within the review group" with the goal of achieving "a balance among various characteristics… [which] include type of organization represented, demographics, experience, and geographic balance" (*National Science Board, 2021a*).

External peer review can occur through one of three methods: ad-hoc only, panel-only, and ad-hoc +panel review. In the ad-hoc only review method, reviewers are sent proposals and asked to submit reviews to NSF through FastLane, NSF's website for proposal submission and review. Panel-only review occurs by soliciting reviews from panelists who convene in person or remotely to discuss their reviews and provide advice as a group to the program officer. Ad-hoc +panel reviews occur using a combination of these processes. Following the review process, NSF program officers review proposal ratings as well as the comments provided by reviewers and panels to make funding recommendations. Program officers also consider other factors, such as the amount of funding available and the award portfolios of their respective programs, prior to making a recommendation to award or decline a proposal.

In addition to program officers, the cognizant division director and/or other NSF officials also oversee the review process. These officials review program officer recommendations before final recommendations are made. Large awards (totaling >2.5% of the awarding directorate's annual budget) are reviewed by the Director's Review Board, or (for awards totaling >1% of the awarding directorate's prior year current plan or >0.1% of NSF's prior year total budget) by the National Science Board.

Once awards are recommended by programs and final division or other programmatic approval is obtained, the recommendation goes to the Division of Grants and Agreements or the Division of Acquisition and Cooperative Support for review of business, financial, and policy implications. Following this review, a final decision is made to award or decline a proposal.

### Internally reviewed proposals
The remainder of proposals belong to special categories that by NSF policy are exempt from external review and may be internally reviewed only. These include EAGER (Early-concept Grants

for Exploratory Research) and RAPID (Grants for Rapid Response Research) awards and proposals submitted through the RAISE (Research Advanced by Interdisciplinary Science and Engineering) mechanism (*National Science Board, 2021a*). Internally reviewed proposals are considered "competitive proposals" by NSF and are thus included within the funding statistics examined in this study.

## NSF merit review criteria

NSF's two major merit review criteria, Intellectual Merit and Broader Impacts, were first adopted in 1998 and revised in 2007 and 2012. The first of these major revisions was undertaken to "promote potentially transformative research" while the second was intended to revise "the elements considered by reviewers" and "articulate the principles upon which the criteria are based" (*National Science Board, 2021a*).

All proposals reviewed by NSF undergo an evaluation for Intellectual Merit and Broader Impacts. Individual Programs may have additional review criteria by which submissions are rated that are specific to the goals of the program. Review criteria are announced in program solicitations and available to PIs when they make their submissions. Typical NSF proposals are reviewed by 3–5 reviewers, whether internally or externally; the number depends on submission mechanism used and individual proposal details. Reviewers are chosen for their expertise and ability to add additional viewpoints to the decision-making process.

Intellectual Merit is intended to assess the potential of the proposal to advance knowledge. Broader Impacts is intended to address "the potential of the proposal to benefit society and contribute to the achievement of specific, desired societal outcomes" (*National Science Board, 2021a*).

## Merit review scores from external reviews

Ad-hoc reviewers provide written reviews that describe the strengths and weaknesses of proposals in the context of review criteria, and rate proposals on a scale from "Poor" (1/5 points on a numerical scale) to "Excellent" (5/5 points). Over the past decade, merit review scores appear to have become more stringent, with proposals receiving a Very Good or higher (>4) dropping ~5% (*National Science Board, 2021a*). The number of highly-rated proposals also declined for URM PIs over this interval by 3–8% (*National Science Board, 2021a*).

NSF notes that "declined proposals represent a rich portfolio of unfunded opportunities—proposals that, if funded, may have produced substantial research and education benefits" (*National Science Board, 2021a*).

## Appendix 2

### Additional context for racial and ethnic categories

Because race is a social construct and racial identity is highly mutable and context-dependent, racial identity is not equivalent to racial identification. While the two are related, racial identification refers to how people classify and designate themselves on surveys and censuses, while racial identity is much more complex. We caution that the racial and ethnic data used in this analysis should be considered to reflect each PIs' identification rather than their true racial identity. Below we describe some caveats to interpreting data on racial identification, as well as clarifications on the naming conventions we employ for specific racial terms and descriptions of racial categorizations.

#### Data for white PIs

Funding outcomes for white PIs are only available 1999–2019 (*Figure 6—figure supplement 1*). As a result, the majority of the analyses for this study cover the interval beginning in 1999, in order to allow for comparison of non-white PI funding rates to funding rates for white PIs. We note that NSF defines white as including persons from the Middle East or North Africa. Just as data aggregation impacts Asian cultural groups (e.g., Hmong, Bhutanese, Vietnamese, Filipino, *et cetera*) who experience racialization and marginalization more strongly than other Asian cultural groups, Middle Eastern (or, more precisely, Southwest Asian) and North African cultural groups experience racialization and marginalization in a distinct way from white European cultural groups (*Awad et al., 2019*). Americans of Southwest Asian and North African (SWANA) descent often overwhelmingly self-identify as an ethnic minority (*Awad et al., 2021*). In addition, SWANA Americans often experience distinct and heightened amounts of historical trauma, pervasive institutional discrimination, and a hostile context within the United States (*Awad et al., 2019*). SWANA data should be disaggregated and reported separately from the white category to reflect these differences. The authors regret that, due to federal standards for collecting race and ethnicity information, racial disparities in NSF funding for SWANA PIs cannot be explored.

#### Data for multiracial PIs

Multiracial identity, like all racial identities, is mutable through time. Those who identify as multiracial for NSF reporting purposes may, in a different social context, identify as monoracial (*Mihoko Doyle and Kao, 2007*; *Liebler et al., 2017*; *Harris and Sim, 2002*). While multiracial identity reporting allows for added nuance in demographic analyses, the nature of multiracial identity adds another layer of complexity to demographic data. When reporting race, many individuals from multiple races will choose to report a single racial identity (*Bratter, 2018*). While in some cases, this behavior is because individuals more strongly identify with one racial group, other cases may stem from fear that inclusion in the multiracial category will result in invisibilization or erasure of the individual's multiple identities (*Gullickson and Morning, 2011*). It is clear that there are numerous challenges involved in the theorization of multiracial identity (*Rockquemore et al., 2009*) and in the definition, inclusion, and interpretation of multiracial data in US demographic analyses (*Charmaraman et al., 2014*), none of which have been satisfactorily resolved. As a result, the NSF's data capture "*a* multiracial population, not *the* multiracial population" of PIs who have submitted NSF proposals (*Harris and Sim, 2002*).

#### Data on underrepresented racial and ethnic minority PIs

Since at least 1990, NSF has defined a category for underrepresented racial and ethnic minorities (URMs) and reported on the proposals, awards, and funding rates for proposals by PIs in this category. While the name of this category changed from "Minority" to "Underrepresented Minority" in 2015, its definition has remained constant since 1990, making the data associated with URMs the most continuous and internally self-consistent demographic information available. NSF defines racial and ethnic URMs to include PIs who are American Indian/Alaska Native, Black/African American, Hispanic or Latino, and Native Hawaiian/Pacific Islander, but to exclude PIs that are white Non-Hispanic or Asian. *Figure 1—figure supplement 3* shows the absolute and relative funding rate data for proposals by PIs who are racial and ethnic URMs in comparison to other available data.

#### Erasure through non-reporting of data and data aggregation

We note that non-reporting for PIs who self-identify as AI/AN and NH/PI—i.e., the major Indigenous groups in the United States—as well as for non-white Hispanic PIs (another group with

ties to Indigeneity in the US), is in itself an act of erasure that has compounding effects on the understanding and mitigation of racial disparities that affect Indigenous groups (*Bratter, 2018*). While the collection of Indigenous data requires care, nuance, and adherence to the data sovereignty protocols of each represented Tribe, Nation, or People (*Carroll et al., 2020*), the current lack of intersectional demographic information available to Indigenous PIs contributes to under- or non-reporting of trends for Indigenous people in STEM. To date, few census questionnaires worldwide enumerate Indigenous peoples, and those that do typically homogenize these groups under broad categorizations (e.g., AI/AN, NH/PI) or classify Indigenous respondents as minorities rather than as distinct peoples (*Peters, 2011*). These practices contribute to the challenges that Indigenous peoples face to document their existence and contribute to continued ignorance of the distinct issues that Indigenous peoples face in STEM.

We are encouraged that the NSF Committee on Equal Opportunities in Science and Engineering (CEOSE) recently recognized in their 2019–2020 Biennial Report the importance of tracking statistics for groups with low representation. In this report, they noted: "small numbers cannot be a rationale to stall progress. Concluding that little can be said with limited data renders underrepresented groups more invisible and creates a roadblock to meaningful changes. To create lasting and impactful changes, organizations should be willing to analyze small numbers, gather detailed interview data on employee experiences, engage managers as allies for changes, and hold themselves accountable to making small numbers grow" (*CEOSE, 2020*). CEOSE encourages the development of innovative strategies and approaches to define, monitor and report success in broadening participation at NSF and across STEM in order to address "challenges related to little or no analyses due to data quality and sample size" (*CEOSE, 2020*).

