## [Decision Letter]

**Decision letter after peer review:**

Thank you for submitting your article "Decades of systemic racial disparities in funding rates at the National Science Foundation" to *eLife* for consideration as a Feature Article. Your article has been reviewed by three peer reviewers, and the evaluation has been overseen by myself. The following individuals involved in review of your submission have agreed to reveal their identity: Nicholas Gilpin; Rachel Bernard.

The reviewers and editors have discussed the reviews and we have drafted this decision letter to help you prepare a revised submission.

Please also note that *eLife* has various formatting requirements about appendixes and supplementary materials (eg figures are not allowed in supplementary materials). I will email you about this in the next few days.

Summary:

This manuscript is extremely well-written and organized. The figures are beautiful (though dense) and the amount of data and information presented is impressive, to say the least. Beyond the strengths in presentation, this study is critical and necessary. Even though this data is technically publicly available, this type of straightforward presentation of NSF funding trends simply does not exist – not from NSF and not in the peer reviewed literature. Past studies have occasionally looked at trends within single disciplines, but this massive dataset reveals systemic problems within a major source of funding for basic research in the United States. And as the authors point out in their discussion, because these trends exist in other funding agencies as well, they are indicative of systemic issues that underpin all of science and society.

The findings of this study will likely have a significant impact in several areas: first, at NSF, where the vast majority of program officers want to have a fair and unbiased proposal review process and a diverse portfolio of awards; and second, among PIs who are not only affected by these discrepancies (particularly non-white PIs) but are unconsciously contributing to these discrepancies through the external review process. I see this paper as being a catalyst for change and an invaluable piece of evidence for those within science who are already fighting for it.

There were several times in this article that I was impressed with the authors' deep consideration of the data (i.e., pointing out paradoxes and surprising trends) and insight into issues I have not seen addressed in the many diversity-focused articles and essays that have come out in the past couple years. In particular, the discussion and analysis around the low funding success of Asian PIs was thoughtful and will likely be surprising to many. Though not considered an "underrepresented minority" by NSF, the authors show clear evidence that this group experiences the largest disparity in funding rates amongst all non-white racial groups.

However, there are a number of points that need to be addressed to make the article suitable for publication.

Essential revisions:

1. Given that Figure 1 seems to be the only figure includes data and analysis prior to 2013 (Figure 5 only has 2015 and 2016 data), the use of the term "decades" is not always justified and should be replaced in a number of places in the article. For example, "Systemic racial disparities in funding rates at the National Science Foundation" might be a more appropriate title for the article.

2. Figure 1 (A): Why the y-axis on the right side has the label "overall number of competitive proposals"? What is the definition of a "competitive proposal"? Does the definition align with NSF's categorization of "competitive proposals" after peer review? If so, does this mean that those submitted proposals that were not considered as being "competitive" after NSF peer review was not included in the analysis?

3. NSF projects may have multiple PIs in case of collaborative projects across institutions, which I believe is becoming more and more common. How were multi-PI projects handled?

4. On a related note, a PI might be awarded multiple projects, in one year or across many. Previous studies suggest that there is a Matthew Effect in getting funding. Because the current analysis does not seem to differentiate individual PIs, I wonder how much multi-project PIs contribute to the existing race disparities we observe.

5. This study seems to overlook another important step in the NSF grant award process – the panel discussion. For many programs in NSF, once proposals are back with external reviews, they are usually evaluated by a panel to discuss their "competitiveness level". Then the program directors decide on the awards based on recommendations from the panel. Is panel discussion data available? What Figure 5B seems to suggest is that the White seems to have the lowest success rate if given the same external review score for 2015? 2016 seems to be slightly different? What about other years? Given the external review seems to favor White consistently (Figure 5A), what does this indicate about the panel process?

6. Discretionary funding decisions are discussed (the good and the bad), but the potential role of grant review panel demographics is not discussed, nor is the process of grant reviewer selection/access. This is especially perplexing in that sub fields with more URM representation actually have worse grant outcomes. Why aren't the higher percentages of URM scientists in those sub fields reviewing grants at higher rates than fields where they are less represented? Or are they?

7. As mentioned in the Discussion, the Asian PI disparity at NSF is very different from what has been reported at NIH. Do the authors have any thoughts as to why that might be?

8. This is alluded to in the Discussion, but it is probably worth directly stating/emphasizing that re-distribution of surplus white awards (just some of them, not even necessarily all of them) would easily bring funding rates for all other racial groups to the overall mean due to the disproportionately high percentage of application that are submitted by white PIs relative to other racial groups.

9. I suggest emphasizing the FOLD advantage (1.7-1.8x) whenever possible to emphasize the magnitude of the disparity.

10. If I am reading Figure 3 correctly, then not only is there an Asian disadvantage relative to white PIs, but also Asian PIs are the only ones that show a major gap between the proposal disparity and the award disparity. This deserves discussion.

11. Within directorates, numbers look similar over time and award type, but what explains the discrepancies in racial distributions/inequities between directorates/disciplines? Is this similar to "topic choice" at the NIH in Hoppe et al. 2019?

12. I found the term "unfunded awards" to be very confusing. I know what it means, but when used in context (like in the sentence, "…BIO contributed 9% of all unfunded Research awards to Asian PIs…" on page 9) it sounds like it refers to proposals that were given awards but no money. I believe that different terminology (for example "funding deficit" or "award deficit") would be clearer and easier to understand.

13. While reading the article I found myself waiting for a section exploring *why*. As in, why are proposals from non-White PIs scored lower?

---

## [Author Response]

Essential revisions:1. Given that Figure 1 seems to be the only figure includes data and analysis prior to 2013 (Figure 5 only has 2015 and 2016 data), the use of the term "decades" is not always justified and should be replaced in a number of places in the article. For example, "Systemic racial disparities in funding rates at the National Science Foundation" might be a more appropriate title for the article.

As suggested, we have removed the term “decades of” from the paper title, and have also removed its use elsewhere in the article, where appropriate.

2. Figure 1 (A): Why the y-axis on the right side has the label "overall number of competitive proposals"? What is the definition of a "competitive proposal"? Does the definition align with NSF's categorization of "competitive proposals" after peer review? If so, does this mean that those submitted proposals that were not considered as being "competitive" after NSF peer review was not included in the analysis?

In the merit review reports from which our data are sourced, NSF uses several terms to describe various categories of proposals, including the phrase “competitive proposals.” Competitive proposals refer to all proposals for which an award or decline decision was made. This category of proposals does not include proposals which were withdrawn by PIs or returned without review by program officers for not meeting certain requirements (e.g., ineligible proposals or PIs; incompliance with solicitation requirements or other policies and procedures). According to the Fiscal Year 2020 NSF Merit Review Report, typically, 1–3% of submitted proposals are returned without review.

Withdrawn proposals or proposals returned without review are not included in any of the reported funding statistics disaggregated by PI demographics or by other characteristics contained within the merit review reports, and are thus not considered in this study. For simplicity and ease of communication, we have opted to drop the word “competitive” when describing the set of proposals for which an award or decline decision was made.

In response to this comment, we have (1) removed instances of the word “competitive” in figures and the main text that refer to all proposals for which an award or decline decision was made, and (2) added several sentences to the Methods and Appendix 1 about “competitive proposals” as a category when describing the NSF merit review process.

3. NSF projects may have multiple PIs in case of collaborative projects across institutions, which I believe is becoming more and more common. How were multi-PI projects handled?

We appreciate this question and suspect many others reading our paper may have the same question as well. We have therefore added our answer to this question about the treatment of proposals by multiple PIs in the NSF merit review reports as a subsection in the Methods called, “Treatment of proposals by multiple PIs in merit review reports.” This section reads as follows:

“When PIs respond to a solicitation for proposals at NSF, PIs may submit a collaborative proposal with multiple PIs or a non-collaborative proposal as a single PI. In the case of collaborative proposals with multiple PIs, these may be submitted in one of two ways: as a single proposal or as multiple proposals submitted simultaneously from different organizations.

“For a submission of a collaborative proposal via the single proposal method, a single PI assumes primary responsibility for the administration of the award and communication with NSF. Other involved PIs from other organizations are designated as co-PIs. If an award is given, only a single award is made. In this case, the demographic information of the PI with primary administrative responsibility is associated with the single proposal and its funding outcome.

“For a submission of a collaborative proposal via simultaneous submissions from multiple organizations, separate proposal submissions are made by each PI, and the project title must begin with the words, “Collaborative Research.” If the collaborative proposal is funded, each organization receives and is responsible for a separate award. In this situation, the demographic information of the PI with primary administrative responsibility for each proposal is attached to the proposal and its funding outcome. For example, if a collaborative proposal submission involves three proposals from three different PIs at different institutions, and this collaborative proposal is funded, then three proposals and three awards with the demographic information of three different PIs are included in the data tables found in the merit review reports.

“Although determining the impact of collaborative research on racial funding rate disparities is of interest, currently available data do not allow for a differentiation between proposals submitted as collaborative work with multiple PIs or proposals by single PIs.”

4. On a related note, a PI might be awarded multiple projects, in one year or across many. Previous studies suggest that there is a Matthew Effect in getting funding. Because the current analysis does not seem to differentiate individual PIs, I wonder how much multi-project PIs contribute to the existing race disparities we observe.

We appreciate this comment and have also wondered the same. Unfortunately, currently available data on funding outcomes by PI race and ethnicity are only reported in terms of proposals and awards, and do not allow us to examine disparities on a per-PI basis. However, there exist data that showcase the prevalence of PIs with multiple active Research awards (Data S12), which we have illustrated in Figure 6—figure supplement 6 and briefly discussed in the “Limitations of data.” For every three-year window since 1995, 35–44% of all PIs who applied for at least one Research grant received at least one award. Of these successfully funded PIs, 13–16% received two awards, 3% received three awards, and 1–2% received four or more awards (Data S12). We also note that the average number of Research awards to successfully funded PIs has been increasing over the past two decades (though this trend is not monotonic).

Suffice to say, we too are curious about the demographic breakdown of these aforementioned award patterns by PI race and ethnicity, and we hope that NSF makes this information available.

5. This study seems to overlook another important step in the NSF grant award process – the panel discussion. For many programs in NSF, once proposals are back with external reviews, they are usually evaluated by a panel to discuss their "competitiveness level". Then the program directors decide on the awards based on recommendations from the panel. Is panel discussion data available? What Figure 5B seems to suggest is that the White seems to have the lowest success rate if given the same external review score for 2015? 2016 seems to be slightly different? What about other years? Given the external review seems to favor White consistently (Figure 5A), what does this indicate about the panel process?

Unfortunately, panel discussion data are not available, and information about external review scores and success rates by review score are only available for 2015 and 2016. However, comparing the external review scores and success rates by score indicates that panel recommendations or discretionary decision-making by program directors, or both, do have an impact on final award or decline decisions.

In response to this comment, we have (1) added an additional description of the proposal success rates by score for white PIs in the subsection, “Racial disparities in external review scores,” and (2) added a paragraph that describes the potential impact of panel discussions on final funding outcomes at the end of the subsection, “Over twenty years of racial funding disparities at NSF, NIH, and other funding bodies.” (We note that this paragraph was also added in response to a later question comparing differences in NIH and NSF outcomes for Asian PIs.)

6. Discretionary funding decisions are discussed (the good and the bad), but the potential role of grant review panel demographics is not discussed, nor is the process of grant reviewer selection/access. This is especially perplexing in that sub fields with more URM representation actually have worse grant outcomes. Why aren't the higher percentages of URM scientists in those sub fields reviewing grants at higher rates than fields where they are less represented? Or are they?

We appreciate this question and agree with this comment. Unfortunately, data on the demographics of reviewers is incomplete for most years and insufficient for analysis due to extremely low response rates in providing demographic information. Therefore, we are unable to determine if the composition of reviewers or panels reflects the demographics of certain subdisciplines, nor are we able to evaluate how the composition of reviewers or panels influences funding decisions.

However, a few studies have examined the relationship between author and reviewer identity. In response to this comment, we have added the following paragraph to the end of the subsection, “The need for disaggregation and expanded approaches to evaluating demographic progress,” following our discussion of the inverse relationship between the proportion of proposals by Black/AA PIs and Black/AA PI relative funding rates:

“The cause of this inverse trend across directorates requires further examination, but a recent study showing author-reviewer homophily in peer review suggests that reviewer, panel, and program manager demographics may play a role. Preferences for shared characteristics may benefit PI applicants with identity markers that are overrepresented amongst evaluators, producing a feedback loop if prior funding success is a desired qualification for reviewer or panel participation. We note however that some studies have shown an opposite effect, in which women reviewers and panelists exhibit a stronger bias against women applicants, indicating that author-reviewer dynamics are complex and require further study.”

7. As mentioned in the Discussion, the Asian PI disparity at NSF is very different from what has been reported at NIH. Do the authors have any thoughts as to why that might be?

We thank the reviewers for this question, which highlighted an important point that bears emphasis. In response, we have added the following two paragraphs to the Discussion describing differences in funding outcomes for Asian and Black/AA PIs at NSF and NIH:

“Future work that leverages differences and similarities in review processes and funding outcomes between NSF, NIH, NASA, and other funding bodies may yield insights on causal mechanisms and offer potential solutions. For example, at face value, Asian PIs have the lowest overall proposal funding rates of all racial groups at NSF while Black/AA PIs have the lowest rates at NIH. While this observation is not false, the overall funding rate disadvantage for Asian PIs compared to white PIs is similar in magnitude at both agencies: NIH research project grant (RPG) proposals by white PIs were funded 1.3 times more than those by Asian PIs in the 2010–2021 period, compared to 1.4 times more for NSF proposals from 2010 to 2020. In contrast, over the same time periods, the funding rate advantage for white PIs compared to Black/AA PIs was 1.7-fold at NIH compared to 1.2-fold at NSF. In other words, the funding disadvantage experienced by Black/AA PIs compared to white PIs is worse at NIH than at NSF. These NIH and NSF data are not wholly equivalent: NIH RPGs only include grants for research activities, and as previously shown, overall NSF proposal outcomes mask larger disparities for Research proposals. However, in the NSF BIO directorate, which has the most disciplinary overlap with the NIH, Research proposals by white PIs had a 1.5- and 1.2-fold advantage over those by Asian and Black/AA PIs in the 2012–2016 period (Figure 4A). The recurrence of the pattern in the BIO directorate suggests that interagency differences in overall Black/AA funding outcomes may remain even if more equivalent data were available for comparison.

“One explanation for this difference may lie in the panel discussion and decision-making phases of the merit review process. At NSF, success rates based on review scores in 2015 and 2016 indicate that funding decisions partially countered the lower scores of proposals by Black/AA PIs, with a smaller effect on proposals by Asian PIs (Figure 5B). This effect has also been observed in other funding contexts where unequal evaluations by gender were counteracted by panels, leading to gender-equal funding outcomes. In contrast, at NIH, proposals by white PIs are often funded at higher rates than those by Black/AA PIs with comparable scores, a pattern that persists within research topic clusters . Whether the countering effect at NSF is primarily occurring at the panel discussion stage, when NSF program officers issue an award or decline recommendation, or when division directors make a final decision is unknown with currently available information. Nevertheless, differences in the way NIH exercises their prerogative to fund proposals outside of rank order likely contributes to the discrepancy in Black/AA PI funding outcomes between NSF and NIH.”

Relatedly, we have noticed that commentary about our work by the broader scientific community has focused on the funding rate disparities for Asian PIs, often describing them as the worst of all racial groups. This reaction is understandable given the prominence of the trends in overall relative funding rates featured in Figure 1, and is not entirely false, especially when considering their cumulative impact. However, the patterns are more nuanced when data are disaggregated by proposal type and directorate: the funding rate disadvantages for Asian and Black/AA PIs at NSF are similar in magnitude for Research proposals, and at the directorate level, funding rate disadvantages for Black/AA PIs are the worst of all other groups in five of the seven directorates. To expand on this key point, we have added the following lines to the Results section.

In “Racial stratification of award types”:

“In addition, these results suggest that the magnitude of disparities for Asian and Black/AA PIs is more similar when considering only Research proposals: on average for the 2013–2019 period, Research proposals by white PIs had a 1.37- and 1.40-fold funding rate advantage over those by Asian and Black/AA PIs, respectively (Data S14). Thus, disaggregating funding statistics by proposal type reveals a more complete picture of these racial funding disparities and their impacts.”

In “Racial disparities across directorates”:

“Between directorates, the size of disparities and the group with the lowest funding rate varied. For Research proposals with available data, Black/AA PIs had the lowest multi-year average funding rate in CISE, EHR, ENG, SBE, and MPS, whereas Asian PIs had the lowest in BIO and GEO. Considering disparities by each year, while Research proposals by Black/AA PIs were consistently the lowest funded in CISE for every year between 2012 and 2016, the group with the lowest funding rate occasionally changed in other directorates. In comparing the magnitude of disparities for Research proposals across directorates, white PIs experienced the largest Research funding rate advantage over Asian PIs in BIO (1.5-fold advantage, multi-year average) and over Black/AA PIs in SBE (1.7-fold advantage, multi-year average; Data S14).”

8. This is alluded to in the Discussion, but it is probably worth directly stating/emphasizing that re-distribution of surplus white awards (just some of them, not even necessarily all of them) would easily bring funding rates for all other racial groups to the overall mean due to the disproportionately high percentage of application that are submitted by white PIs relative to other racial groups.

We agree that redistributing some of the white award surplus can have a considerable impact on reducing or eliminating funding rate disparities for other racial groups. Because of the large size of the white proposal pool, such redistributions would have a minimal — if not imperceptible — effect on funding rates for white PIs.

For example, in terms of overall award surpluses and deficits in 2019 (Figure 2), if just 9 awards to white PIs were instead made to Black/AA PIs to close the funding gap, the funding rate for proposals by white PIs would decrease by *less than a twentieth of a percent* — from 32.32% to 31.27%. Reducing the Asian award deficit by half with 216 awards from the white award surplus would decrease the funding rate for proposals by white PIs by 1.06%.

Such re-distributions could be achieved through a discretionary decision to redirect awards to white proposals with the lowest review scores to proposals by underfunded PIs with comparable or better scores. Indeed, such a strategy has been proposed to eliminate the gap for Black/AA PIs at the NIH by others (Taffe and Gilpin, 2021). We also wonder about what may be learned once these disparities are considered on a per-PI basis, given that 13–16% of funded PIs have two awards, 3% have three awards, and 1–2% at least four (Data S12; Figure 6—figure supplement 6). The number of PIs with at least four awards may be a significant proportion of or comparable to the total number of PIs belonging to certain racial groups.

While the Asian award deficit is more difficult to eliminate due to its size, we consider this obstacle an artifact of the construction of this racial category. Here, we are reminded of the “Asian or Pacific Islander” category that was once an option for race and ethnicity in the 1990 and 2000 US Census. A 1997 government directive separated this category into “Asian” and “Native Hawaiian and Pacific Islander” options after thousands of Native Hawaiians commented that the combined category was inadequate for data that monitored the conditions of Native Hawaiians and other Pacific Islander groups (US Office of Management and Budget, 1997). The disaggregation of the Asian or Pacific Islander category into separate groups allowed for data-based justifications of government funding and resources to Native Hawaiians and other Pacific Islander groups which may not have occurred otherwise. In a similar vein, the current Asian category embodies many diverse groups with unique experiences, obstacles, and needs, and the total size of the Asian award deficit alone should not be a deterrent to implementing solutions that address it.

We also note here that care must be taken in treating the award surplus and deficit numbers at face value, especially if using them as an equity metric. We have added the following paragraph to the Methods section to add context:

“As previously discussed, the examination of funding rate disparities by Research proposals and by directorate illustrates the necessity of data disaggregation, as several patterns are hidden within overall funding statistics for all proposals NSF-wide. Similarly, the same is true for award surpluses and deficits. Because each group has different submission patterns in terms of the proportion of Research versus Non-Research proposals and their distribution across directorates, total award surpluses and deficits for each group that are calculated at more granular levels will differ from those calculated at broader levels. For example, for the 2019 fiscal year, if we calculate the total award surplus or deficit for white, Asian, and Black/AA PIs by separately calculating the award surplus or deficit for Research and Non-Research proposals and then summing these values, we arrive at totals of +807 for white (+637 Research, +171 Non-Research; note rounded values), -364 for Asian (-369 Research, +4 Non-Research), and -18 for Black/AA PIs (-17 Research, -1 Non-Research). These values differ from those calculated using overall funding rates for all proposals: +798 for white, -432 for Asian, and -4 for Black/AA PIs (Figure 2). As a similar exercise, for the 2015 fiscal year, if we calculate the total Research award surplus for white PIs by summing the award surplus from each directorate and office (Data S4), we arrive at a total Research award surplus of +393 compared to +447 if calculated based on overall funding rates for all Research proposals. Award surpluses and deficits should be calculated at more granular organizational levels (i.e., divisions and programs) and proposal types (e.g., CAREER grants), but such data are not currently available.”

9. I suggest emphasizing the FOLD advantage (1.7-1.8x) whenever possible to emphasize the magnitude of the disparity.

We appreciate this suggestion. In sections where we compare funding outcomes of a particular group with those of white PIs, we have used the “1.7­–1.8x” or “1.7–1.8-fold” language to emphasize the magnitude of disparities (for example, in the new paragraphs comparing Asian and Black/AA PI funding outcomes between NSF and NIH). We have also added another collated dataset to our online data repository that calculates funding disparities in terms of the white PI funding rate advantage or disadvantage compared to other racial groups for all proposals, all Research and Non-Research proposals, and Research and Non-Research proposals by directorate (Data S14).

10. If I am reading Figure 3 correctly, then not only is there an Asian disadvantage relative to white PIs, but also Asian PIs are the only ones that show a major gap between the proposal disparity and the award disparity. This deserves discussion.

We wish to respond to this comment, but we are not sure as to what is meant by “proposal disparity” and “award disparity.” Based on this comment and another minor comment from below, our takeaway is that Figure 3’s presentation of data must be improved for reader comprehension.

To this end, we have modified this figure and edited the caption for clarity. In the bar charts, we no longer show a light-colored bar representing the proportion of proposals or awards for Non-Research. Instead, the only bars shown represent Research proportions by group, which can then be visually compared to Research proportions for all groups combined that are represented by the horizontal lines. We are reminded of data visualization expert Edward Tufte’s guiding principle to maximize the “data-ink ratio” and believe these changes better communicate the key takeaway of Figure 3—that racial funding disparities are greater for Research proposals, which contributes to a stratification in Research versus Non-Research activities by race (e.g., only 46–63% of awards to Black/AA PIs are for Research, far below overall proportions of 71–76%).

11. Within directorates, numbers look similar over time and award type, but what explains the discrepancies in racial distributions/inequities between directorates/disciplines? Is this similar to "topic choice" at the NIH in Hoppe et al. 2019?

We appreciate this question. Currently available information limits our ability to understand the drivers behind the magnitude of disparities across directorates. As mentioned previously, the inverse relationship found across directorates between the proportion of proposals by Black/AA PIs and the relative funding rate for proposals by Black/AA PIs is compelling for this reason.

Hoppe et al. (2019) also found that Black/AA PIs experience both overall and within-topic proposal funding rate disadvantages (see Table 1 and Figure S6 in Hoppe et al., 2019), so our observation that NSF Research proposals by Black/AA PIs have negative funding rates in nearly every directorate is consistent with those findings. Unfortunately, no paper or report thus far has provided information on NIH funding rates by PI race and ethnicity that are disaggregated by the 24 grant-issuing institutes and centers (ICs), the counterparts to NSF directorates in that they manage all award decisions. These data should be made available to allow for the comparison of NSF and NIH funding outcomes using more equivalent data types.

In response to this comment, we have added the following line to a paragraph in the subsection, “Over twenty years of racial funding disparities at NSF, NIH, and other funding bodies”: “Our finding that racial disparities persist at the directorate level is consistent with an NIH study showing that Black/AA PIs experience both overall and within-topic funding rate disadvantages compared to white PIs.”

12. I found the term "unfunded awards" to be very confusing. I know what it means, but when used in context (like in the sentence, "…BIO contributed 9% of all unfunded Research awards to Asian PIs…" on page 9) it sounds like it refers to proposals that were given awards but no money. I believe that different terminology (for example "funding deficit" or "award deficit") would be clearer and easier to understand.

We appreciate this comment and suggestion. In response, we have changed our terminology to “award surplus” and “award deficit.”

13. While reading the article I found myself waiting for a section exploring *why*. As in, why are proposals from non-White PIs scored lower?

We appreciate this comment. Without more detailed information and data about reviewer feedback, review scores beyond just averages, proposal content, and more, we cannot assign specific cause(s) for the differences in review scores between groups. Answering this question in the NIH context has been the source of several papers on its own (e.g., Hoppe et al., 2019).

However, we explore this question throughout the subsection called, “Decades of cumulative advantage and disadvantage at the NSF.” We mention the “Matthew effect” of cumulative advantage, a reinforcement effect in which previous funding success enables greater success in subsequent attempts to acquire funding. This phenomenon may be especially relevant in merit review processes that explicitly consider prior productivity or funding success in evaluations, which is the case in evaluations of NSF proposals (e.g., the required “Results from Prior NSF Support” section). Such requirements may disadvantage non-white PIs if non-white PIs have a more difficult route to acquiring funding. Similarly, we also include a paragraph about other feedback loops that influence and are influenced by these racially disparate funding outcomes:

“These trends represent just one facet of the series of interdependent systems in STEM that manifest unequal outcomes. A litany of prior work shows that while PIs of certain dominant or majority groups benefit from a system of cumulative advantage, particularly white men at elite institutions, those of underrepresented or historically excluded groups are systematically burdened with barriers at every stage of their professional development — from placement into lower-prestige institutions as faculty, smaller institutional start-up funds, smaller and less beneficial collaboration networks, disproportionate service expectations, lower salaries, increased scrutiny and tokenization, and added stressors in suboptimal work environments, to gaps in citations, publications, promotions, and peer recognition that increase with career stage. Together, these barriers traumatize researchers, aggravate attrition, and impair health. The synthesis of these interlocking dynamics magnifies and perpetuates a cycle of funding disadvantage for marginalized researchers, **functioning as both a cause and effect of the racial funding disparities described herein.**”

We have also described in the “Examining the culture of meritocracy” section the many ways in which proposals by non-white PIs may be scored lower, such as the devaluation of ideas and topics studied by non-white groups.

*[Author response to previous editorial comments from other journals]*

In the spirit of *eLife*’s open and transparent peer review process, we, the authors, would also like to share with the scientific community the editorial feedback from two journals that received an earlier version of this manuscript for consideration, but declined to send the paper out for peer review. We include here our responses and the ways in which this feedback improved our paper.

Editorial comment from Journal #1:The analysis is mainly descriptive. It would also have been more useful if numerous factors such as academic rank and place of employment (as determinants of institutional support system) were also taken into account since they are crucial determinants of overall disparity.

We wholly agree that investigating the influence of academic rank and institution on funding outcomes is critical and likely plays a significant role in these racial disparities. Unfortunately, such data have not been made publicly available by NSF. During our analysis, we submitted a formal request to NSF for such information, but this request was not fulfilled. This lack of data availability hampers efforts to examine these disparities beyond a descriptive analysis. Despite these limitations, we are still able to observe major patterns that are consistent at multiple organizational levels at NSF and across multiple STEM funding bodies, which alone have yielded insights on potential mechanisms that influence these racially disparate outcomes (e.g., the comparison of NSF and NIH funding outcomes suggests that discretionary decision making by panels/study sections and/or program officers influences the magnitude of disparities). As yet, a description of these widespread patterns has not been made elsewhere in the published literature or in official reports.

In response to this feedback, we have expanded the “Limitations of data” section in our manuscript to further clarify that additional data on academic rank and institution are currently unavailable but needed for future work to contextualize these racial funding disparities.

Editorial comment from Journal #2:This is a study of racial disparities in the funding rates of grants awarded by the National Science Foundation. The authors find that white principal investigators (PIs) are funded at higher rates than most non-White PIs (including Black/African American, Hispanic, and Asian PIs). The authors strongly suggest with their language (e.g., "pervasive racialized disparities", "systematic racialized biases") that these disparities are caused by racial discrimination and biases against non-white PIs. This conclusion is unwarranted given the limited data that the authors have, and given the lack of research design. This is a clear case of confusing correlation with causation, and as such does not meet the bar that we expect at <REDACTED>. Moreover, even judging the paper only on the merits of its descriptive qualities, the evidence that the authors present is too limited. Without detailed information on the quality of the proposals, the qualifications of the PIs, the PI's institution, the topic, etc., in addition to race and ethnicity, the descriptive analysis is too restricted and does not allow for a deep descriptive study of this question.

We disagree with the editor’s characterization of our paper, which reduces our study to an analysis that invokes “racial discrimination and biases against non-white PIs” as the sole cause of these racial disparities. Based on the overall context of these comments, we infer that when the editor says, “racial discrimination and biases,” they are referring to racial bias exhibited by individual actors within the proposal review process. This comment is interesting from an ethnographic and social psychology perspective, in that similar responses were observed in the aftermath of the Ginther et al. (2011) paper that first showed racial disparities in NIH funding (Stemwedel, 2016; Kington and Ginther, 2018). As a result of such responses, subsequent research on racial gaps in NIH funding focused on answering one question: “Are reviewers, panels, or other individuals involved in the NIH proposal review process racially biased?” When this question was indirectly addressed by finding other factors like topic choice or publication productivity to “explain” part (but not all) of the racial funding gaps, or when individual reviewer bias was deemed difficult, if not impossible to measure in observational settings with current methods, progress on addressing the racial funding gaps at NIH stalled. "The long and the short of it... was that these roadblocks to NIH amassing detailed data on the causal processes responsible for the racial disparity in grant funding were also roadblocks to NIH taking any further steps to address the funding disparity" (Stemwedel, 2016).

The last decade of NIH’s research and policy changes demonstrates that while understanding the causal processes behind these racial funding disparities is both important and of natural interest to many (including ourselves, the authors), it is crucial that these racial funding gaps be considered within the broader societal context of structural and systemic racism. Here, we are reminded of a social phenomenon described by social psychologists Julian Rucker and Jennifer Richeson in their 2021 recent paper, “Toward an understanding of structural racism: Implications for criminal justice” in *Science* (Rucker and Richeson, 2021): “Beliefs about the nature of racism—as being solely due to prejudiced individuals rather than structural factors that disadvantage marginalized groups—work to uphold racial stratification. … Absent an appreciation of structural racism, white Americans are likely to interpret evidence of racial disparities in incarceration through the lens of stereotypes about racial minority criminality, thereby justifying the inequality. … Because structural racism is characterized by policies, practices, and/or laws that have a disparate impact on members of particular racial or ethnic groups, evidence of racially disparate outcomes is the first indicator of its operation.” In a retrospective of responses to the NIH racial disparities, both by the NIH and the broader biomedical community, we observe many aspects of these social phenomena. Thus, following the precedent set by Rucker, Richeson and other prominent scholars of social inequality, in our approach to analyzing racial funding disparities at NSF, we foreground the structural nature of systemic racism.

Furthermore, even if approaching these racial funding disparities with the traditional deficit-oriented framing that has largely fallen out of favor in higher education research, we push back on the notion that all other explanations for racial funding disparities must be exhausted, in which the “null” or “a priori hypothesis” is that systemic racism does not influence institutions of science (Dzirasa, 2020). Given the historical and social context in which the NSF and many other STEM organizations emerged, as sociologist Victor Ray writes, “It is safer, and likely more realistic, to start with the assumption that organizations are contributing to racial inequality unless the data shows otherwise” (Ray, 2019b).

In response to this feedback, we have (1) expanded the “Limitations of data” section in our manuscript to further clarify that additional data on other factors like career stage, institution, topic, *et cetera*, are needed to further contextualize these patterns; (2) added a paragraph to the “Examining the culture of meritocracy“ section to state how we define and use “systemic racism” in this paper; and (3) added text throughout the Conclusions section to further emphasize the importance of not reducing this study as a result “for” or “against” reviewer bias, which misses the forest for the trees, failing to appreciate the larger problem.